# New type of evidence for secondary ice formation at around -15 °C in mixed-phase clouds

Claudia Mignani[1], Jessie M. Creamean[2,3*], Lukas Zimmermann[1], Christine Alewell[1], Franz Conen[1]

[1]Institute of Environmental Geosciences, University of Basel, Basel, 4056, Switzerland
[2]Cooperative Institute for Research in Environmental Sciences, University of Colorado, Boulder, CO 80309, USA
[3]Physical Sciences Division, National Oceanic and Atmospheric Administration, Boulder, CO 80305, USA
*Now at: Department of Atmospheric Science, Colorado State University, Fort Collins, CO 80521, USA

*Correspondence to*: Claudia Mignani (claudia.mignani@unibas.ch) or Franz Conen (franz.conen@unibas.ch)

**Abstract.** Ice crystal numbers can exceed the numbers of ice-nucleating particles (INPs) observed in mixed-phase clouds (MPCs) by several orders of magnitude also at temperatures that are colder than -8 °C. This disparity provides circumstantial evidence of secondary ice formation also other than via the Hallett-Mossop process. In a new approach, we made use of the fact that planar, branched ice crystals (e.g. dendrites) grow within a relatively narrow temperature range (i.e. -12 °C to -17 °C) and can be analysed individually for INPs using a field-deployable drop freezing assay. The novelty of our approach lies in comparing the growth temperature encoded in the habit of an individual crystal with the activation temperature of the most efficient INP contained within the same crystal to tell whether it may be the result of primary ice formation. In February and March 2018, we analysed a total of 190 dendritic crystals (∼3 mm median size) deposited within MPCs at the High Altitude Research Station Jungfraujoch (3580 m a.s.l.). Overall, one in eight of the analysed crystals contained an INP active at -17 °C or warmer, while the remaining seven most likely resulted from secondary ice formation within the clouds. The ice multiplication factor we observed was small (8), but relatively stable throughout the course of documentation. These measurements show that secondary ice can be observed at temperatures around -15 °C and thus advance our understanding of the extent of secondary ice formation in MPCs, even where the multiplication factor is smaller than 10.

## 1 Introduction

Ice-nucleating particles (INPs) are required to catalyse primary ice formation in clouds at temperatures above -36 °C via heterogeneous freezing (e.g. Vali et al., 2015). In mixed-phase clouds (MPCs), heterogeneous freezing is expected to generate ice crystals, but secondary ice production mechanisms can also enhance the ice crystal number concentration (Cantrell and Heymsfield, 2005). The secondary production of ice particles requires the prior presence of other ice particles (Vali, 1985).

For example, secondary ice crystals can result from rime splinters that are released upon riming (i.e. supercooled cloud droplets that freeze upon contact with a solid hydrometeor) of ice crystals at temperatures between -3 °C and -8 °C (Hallett and Mossop, 1974; Jackson et al., 2018). Other than the well-known Hallett-Mossop process, mechanisms proposed for secondary ice production include ice-ice collisional breakup (e.g. Vardiman, 1978; Phillips et al., 2017), droplet shattering or fragmentation

upon freezing (e.g. Takahashi and Yamashita, 1970; Lauber, 2018) and sublimation fragmentation (e.g. Bacon et al., 1998). These processes and indications for their occurrence in the atmosphere are summarised in Field et al. (2017). Sullivan et al. (2018a) have recently studied three of the above-mentioned secondary ice formation processes in terms of their thermodynamic and primary ice requirements in a parcel model. They showed that INP concentration can be as low as 2 m$^{-3}$ (0.002 L$^{-1}$) to initiate ice multiplication by ice-ice collisional breakup. Furthermore, the number of INPs is less important with regard to

cloud formation than a sufficiently warm cloud base temperature and modest vertical updraft velocity for frozen droplet shattering and rime splintering (Sullivan et al., 2018a). When droplet shattering and ice-ice collisional breakup were implemented into a large-scale weather model, secondary ice contributed as much to the ice crystal number concentration as did primary ice nucleation, even though high ice crystal numbers remain underestimated by the model (Sullivan et al., 2018b).

While modelling studies accounting for secondary ice production can to some extent explain the observed ice crystal numbers (e.g. Sullivan et al., 2018b), field measurements have not been conclusive as to the contribution of secondary ice production mechanisms until present days. Kumai (1951, 1961) and Kumai and Francis (1962) found an insoluble particle of 0.5 to 8 µm in size in the centre of almost every one of the about 1000 snow crystals they collected. The particles they found were clay and related minerals and were assumed to have initiated the formation of the crystals. Bigg (1996) suggested to repeat the

experiments by Kumai and Francis (1962) and to look at the ice nucleation properties of these particles. One reason why it can be misleading to equate ice residuals with INPs is that MPC-generated ice crystals can contain cloud condensation nuclei (CCN) which have been collected upon riming but have not acted as INPs. One possibility to overcome this issue is to sample ice residuals of freshly formed, small ice crystals (< 20 µm), which are assumed to have grown by the initial phase of vapour diffusional growth only (Mertes et al., 2007; Kupiszewski et al., 2015). On mountain-top stations, where such crystals can be

collected in-cloud, however, hoar frost (cloud droplets frozen onto surfaces) can be a strong source of small (i.e. < 100 µm) ice crystals (Lloyd et al., 2015; Farrington et al., 2016; Beck et al., 2018). Hoar frost grows in saturated conditions, breaks off when windy, and broken-off segments can become ingested into clouds and commonly mistaken for secondary ice (Rogers and Vali, 1987). Residuals in hoar frost particles are CCN that had not been activated as INPs. Only droplets freeze upon contact with an iced surface while ice particles bounce off and remain in the airflow, a principle applied in counterflow virtual

impactor inlets used to separate ice from liquid in MPCs (Mertes et al., 2007). Current ice selective inlets are not able to separate primary from secondary ice (Cziczo et al., 2017).

Another possibility to investigate secondary ice is by comparing the concentration of INPs with that of ice crystals in the same cloud. Most such studies report large discrepancies between measured INPs and ice crystal numbers (e.g. Hobbs and Rangno,

1985; Lasher-Trapp et al., 2016; Ladino et al., 2017; Beck et al., 2018) the latter being several orders of magnitudes higher than the former. To the contrary, a good agreement between INPs and ice crystals was found by Eidhammer et al. (2010) in an orographic wave cloud. Furthermore, INP concentrations from bulk precipitation samples cannot be disentangled to the level of individual hydrometeors (Petters and Wright, 2015). Riming can affect the INP spectrum of a bulk precipitation sample by

adding scavenged INPs immersed in supercooled cloud droplets, which have not been activated under *in situ* conditions (Creamean et al., 2018b). Further, ice-nucleation active microbes can be scavenged by raindrops below cloud and alter the spectrum (Hanlon et al., 2017).

Another way to separate primary from secondary ice particles could be INP assays on individual hydrometeors collected within

MPCs. The first experiment in which individual hydrometeors were analysed for INPs, and the only one to our knowledge, was conducted by Hoffer and Braham (1962). The hydrometeors they had collected from aircraft were large, frozen water drops that had grown through riming ("snow pellets" or "ice pellets"; Braham, 1964) within summer clouds. Because they all (n = 301) re-froze only at temperatures substantially lower than the estimated cloud top temperature, the authors presumed them to be of secondary origin. However, an ice multiplication factor (i.e. the number of all ice particles divided by the primary

ice particles) could not be estimated because the number of primary ice particles was zero.

In this study, similar to the one by Hoffer and Braham (1962), we collected in-cloud hydrometeors to obtain *in situ* evidence of secondary ice formation. We concentrated on secondary ice formation at around -15 °C for three reasons. First, the growth habit of ice crystals forming in super-saturated conditions between -12 °C and -17 °C is well and distinctively defined. These

are single, planar, branched, sector-type or dendrite-type habits (Nakaya, 1954; Magono, 1962; Magono and Lee, 1966; Takahashi et al., 1991; Takahashi, 2014; Libbrecht, 2017) that grow by vapour diffusional growth into a diameter of several millimetres during a vertical fall of a few 100 m (Fukuta and Takahashi, 1999). Second, Westbrook and Illingworth (2013) observed a long-lived supercooled cloud layer with a cloud top temperature around -13.5 °C, which continued to precipitate ice crystals well beyond the expected exhaustion of its INP reservoir. Third, laboratory investigations revealed ice-ice collision

to be most effective in producing secondary ice particles at around -16 °C (Takahashi et al., 1995), or in collisions involving dendritic crystals (Griggs and Choularton, 1986). Assuming that the growth temperature of a crystal is not much different from the temperature of its initial formation, these observations suggest that evidence for secondarily formed crystals might be obtained by collecting planar, branched snow crystals from supercooled clouds and testing them individually for the presence of INPs that might have nucleated their formation (i.e. INPs that were activated between -12 °C and -17 °C).

## 2 Experimental

### 2.1 Location and meteorological conditions

Between 15 February and 22 March 2018, we collected and analysed a total of 229 planar, sector- and dendrite-type ice crystals (i.e. ice crystals of a size larger than 1.3 mm in diameter) during cloudy conditions at the High Altitude Research Station

Jungfraujoch (3580 m a.s.l.) in the Swiss Alps. During the collection, cloud base height as measured by MeteoSwiss with a ceilometer located 5 km northwest of Jungfraujoch (Poltera et al., 2017) was on average 950 m below the station (zB, Table 1). Based on air temperature measured by MeteoSwiss at Jungfraujoch, cloud base height and an assumed moist adiabatic lapse rate of 7.5 °C km$^{-1}$ (plausible for approximately 650 hPa and -10 °C) we estimated that daily mean cloud base temperatures (CBT) were between +1 °C and -12 °C. The mean air temperature at the station during the sampling periods was

-11.0 °C (±3.6) and the mean wind velocity was 9.1 m s$^{-1}$ (±3.9). On three days air masses arrived mainly from south-east (SE) or east (E), and on seven days from north-west (NW).

### 2.2 Single crystal selection and analysis

We collected snow crystals on a black aluminium plate (40 cm x 40 cm) at about 1 m above the floor of the main terrace of the Sphinx Observatory at Jungfraujoch and analysed the crystals inside a small, naturally cold (-1 °C to -7 °C) anteroom

between the terrace and the laboratory. Among a usually wide variety of shapes and sizes precipitating onto the plate we selected what we considered to be single, planar, branched or dendritic ice crystals (from here on "dendrites"), which can safely be assumed to have grown within MPCs at temperatures around -15 °C (Nakaya, 1954; Magono, 1962; Magono and Lee, 1966; Takahashi et al., 1991, Takahashi, 2014; Libbrecht, 2017). Generally, we exposed the plate for some seconds to the precipitating cloud until at least two dendritic snow crystals had deposited on it and then analysed those. Our selection criteria

excluded small or irregular ice crystals, which are more typical for hoar frost particles which might have been generated by local surface sources around the station (Llyod et al., 2015; Farrington et al., 2016; Beck et al., 2018). Rime on selected crystals is of little concern in our approach and was accounted for (see Sect. 2.3).

Selected crystals were documented by macro (1:1) photography (camera: OM-D E-M1 Mark II, pixel width: 3.3 µm; objective:

M.Zuiko ED 60mm f2.8; flash: SFT-8; all items from Olympus, Tokyo, Japan) stabilised by a focusing rack (Castel-L, Novoflex, Memmingen, Germany) propped up on the aluminium plate. The size of our crystals was determined by using ImageJ (Rueden, 2017; Schindelin, 2012). Images were later analysed visually and not by machine learning methods, such as developed by Praz et al. (2017), for the habit, including the degree of riming both categorized according to the latest ice crystal classification scheme, as presented by Kikuchi et al. (2013). The scheme catalogues solid precipitation particles into a total of

121 categories and provides for each category a representative image.

After selecting the crystals, we tested them for the most efficient insoluble INP they contain that can be activated through immersion freezing using a custom-built cold-stage (Fig. 1; more details in supplement). A cold-stage is a drop freezing apparatus, on which droplets are deposited onto a cooling surface and the temperature at which they freeze is observed (Vali, 1971a). This technique is commonly used today to assess the activation temperature of INPs immersed in droplets. Observations have shown that an overwhelming majority of ice particles originate from supercooled liquid clouds at temperatures > -27 °C, which strongly suggests that the initial process of ice formation in MPCs > -27 °C occurs through immersion freezing (Westbrook and Illingworth, 2011). The cold-stage used in this study is meant to be taken into the field, can be set up within minutes and operated without additional infrastructure (i.e. no cooling water or lined power is required). It consists of a gold-plated copper disk with a surface diameter of 18 mm, which is large enough to easily accommodate simultaneously two dendrites and two control droplets (roughly 1 cm apart from each other).

With a fine brush, two crystals are transferred onto the cold-stage surface thinly covered with Vaseline® petroleum jelly (Tobo, 2016; Polen et al., 2018) before being analysed within the next minutes (Fig. 1a). The manual application of Vaseline® requires precision and clean gloves in order to get an as uniform and clean cover as possible. At the transfer of the crystals, the surface of the stage was at a temperature between +1 °C and +5 °C, which is a common temperature range to store INPs in water for several hours before analysis (e.g. Wilson et al., 2015). Upon deposition onto the cold-stage the crystals melted into liquid droplets. To aid visual detection of freezing, we increased the size of the melted crystal droplets by adding 3 µL of ultrapure water (Molecular Biology Reagent, Sigma-Aldrich) with a pipette (using a new tip for each measurement run). The melted crystal containing all residuals and potentially the INP that had triggered its formation, has a rather small volume compared to the added water. For each crystal a control droplet (3 µL) of the same ultrapure water was placed next to the melted crystal droplet and served as control (blank) (Fig. 1b). Then we ramped the temperature of the cold-stage down to -25 °C. Shortly after the cold-stage temperature reached a value below the surrounding air temperature, we covered it with a transparent hood to minimise the chance for contamination from the environment surrounding the droplets and to prevent condensation on the cold-stage (Polen et al., 2018). From -12 °C and below we limited the cooling rate to 3 °C min$^{-1}$. The freezing of the droplet and thus the presence of the most efficient INP was detected visually, and the corresponding temperature was recorded manually (Fig. 1c). The presence of an INP active at -17 °C and warmer (INP$_{-17}$) was taken as evidence for the tested dendrite to have been generated through primary ice formation. Nevertheless, extending the drop freeze assay down to -25 °C is useful to determine the fraction of rime associated with single crystals (see Sect. 2.3). In total, the procedure (i.e. collecting and analysing two samples) takes ~15 minutes, a time interval which may allow for a reduction in particle surface area due to coagulation (Emerstic et al., 2015). After a test was complete, we cleaned the cold-stage carefully with isopropanol.

**2.3 Accounting for riming**

A rimed ice crystal has collected liquid cloud droplets, each of them containing a CCN that may cause freezing of the droplet containing the residuals of this crystal. Such a CCN may be activated on the cold-stage as INP (from here on: scavenged INP),

although it had not initiated the formation of the collected dendrite. The median concentration of INPs active at -25 °C or warmer (INP$_{-25}$) was determined for bulk rime samples collected on impactor plates ($conc_{rime}$) and used to estimate the mass of rime associated with a single dendrite ($m$):

$$m = \frac{\ln((1-FF_{crystal})^{-1}))}{conc_{rime}}, \qquad \left[g\ rime\ crystal^{-1} = \frac{INP_{-25}\ crystal^{-1}}{INP_{-25}\ g^{-1}\ bulk\ rime}\right] \qquad (1)$$

with $FF_{crystal}$: the frozen fraction of INP$_{-25}$ in the analysed dendrites (after subtracting the control).

This step was necessary to estimate the contribution of scavenged INP$_{-17}$ representing false positives of primary ice crystals in our results. They were estimated from the average mass of rime associated with a single dendrite (Eq. 1) and the concentration of INP$_{-17}$ within the independent rime samples as described next.

Independent rime samples were collected with a plexiglass impactor plate (Lacher et al., 2017) suspended on the railing of the terrace at Jungfraujoch for a few to several hours (~1-13h). In total, 30 samples of aggregated rime droplets were collected between 15 February and 11 March. The freezing experiments of the rime samples were done with a drop freezing assay similar to the set up described above which was used for the single crystal analysis. However, rime samples were melted and portioned with a sterile syringe into 2.5 µL droplets and analysed with a drop freezing cold plate following the description in

Creamean et al. (2018a). Of each sample 300 droplets were cooled until all droplets were frozen. The cumulative number of INPs active at a certain temperature (with a temperature interval of 0.5 °C) was calculated by taking into account the observed numbers of frozen droplets at a temperature, the total number of droplets and the analysed volume of sample (Vali, 1971b). The main reason for the use of a second cold-stage was to ensure that the custom-built one was always ready for single crystal analysis in case dendrites were precipitating. Other than that, the drop freezing cold plate has a larger surface on which more

droplets can be analysed at a time making it more suitable for rime analysis. However, it also requires an external refrigerated circulation bath, lined power and it is relatively large, making it impossible to put it into the anteroom and to analyse single crystals.

## 3. Results and discussion

Of the 229 crystals analysed in the field 39 had to be excluded retrospectively because a closer inspection of the enlarged

photographs showed that they were either not planar or not branched. Most of the excluded crystals were spatial or radiating assemblages of plane-type crystals (P6 or P7, according to Kikuchi et al. (2013)) and may hence have been initiated at temperatures < -20 °C (Bailey and Hallett, 2009). The remaining 190 crystals were confirmed as planar and branched, i.e. having a habit that typically forms between -12 °C and -17 °C. They had been collected from a pathlength of 2368 km through a large number of MPCs from different wind directions (sum of sampling duration multiplied by average wind speed; see

Table 1). A large fraction of them were rimed (31%) or densely rimed (51%) dendrites (R1c or R2c, according to Kikuchi et

al. (2013); see Fig. S3 for examples); while the remainder belonged to other categories (in order of decreasing frequency: graupel-like snow of hexagonal shape, hexagonal graupel, composite plane-type crystals, dendrite-type crystals, sector-type crystals or R3a, R4a, P4, P3, P2, respectively, according to Kikuchi et al. (2013)). The greatest length in the a-axis (outer diameter) of the 190 crystals ranged from 1.3 to 7.6 mm, with a median of 2.8 mm, a mean of 3.1 mm and a standard deviation

of 1.1 mm.

We found no INP active above -12 °C present in the crystals. In 24 of the 190 crystals an INP active between -12 °C and -17 °C was present (Fig. 2). In the other 166 crystals no INP was found between -12 °C and -17 °C. They either refroze below -17 °C (95) or stayed supercooled until -25 °C (71). The lack of $INP_{-17}$ indicates that the formation of these crystals was most

likely not triggered by heterogeneous freezing, but through a secondary ice formation process. It is highly unlikely that these crystals had grown from homogenously frozen cloud droplets. Homogenous freezing at a temperature well below -20 °C results in a polycrystalline initial ice crystal from which a polycrystalline snow crystal develops (Furukawa, 1982), and not a single crystal like a dendrite. Blanks that froze above -17 °C were limited to one count, occurring between -16 °C and -17 °C (not accounted for in further analysis). Between -17 °C and -25°C, 40 control droplets froze; the rest (149) stayed supercooled until

-25 °C. A frozen fraction of 21% of the control droplets at -25 °C is a rather low fraction compared to the results with pure water droplets (1 µL) on a Vaseline-coated substrate presented recently by Polen et al. (2018).

Throughout the observation period of 10 days the daily fraction of primarily nucleated ice was relatively stable (Fig. 3). From these results, we conclude that about one in eight of the analysed (24/190) planar, branched crystals resulted from primary ice

formation. Seven of eight were likely generated through a process of secondary ice formation given they did not refreeze above -17 °C. The uncertainty associated with the number of primary crystals in our observations is about 20% ($\sqrt{24}/24$). Since we have randomly sampled crystals from many clouds over a prolonged period, we can extrapolate the found multiplication factor to dendrites in MPCs at Jungfraujoch during winter months in 2018 but we can not make detailed judgements about single clouds.


Our preliminary conclusion is based on the following four assumptions: The first assumption is that INPs embedded in natural ice crystals can be repeatedly activated at the same temperature. Second, that the analysed crystals did not grow from aerosolised parts of hoar frost growing on surrounding surfaces (Lloyd et al., 2015; Farrington et al., 2016; Beck et al., 2018). Third, that initial ice formation leading to the growth of the analysed crystals likely did not occur at a temperature colder than

-17 °C. And, fourth, that the detected $INP_{-17}$ were not scavenged during riming of a secondarily formed crystal.

We are confident that the first condition (i.e. that INPs are stable over many refreezing cycles) for our preliminary conclusion is met. Although substantial fractions of bacterial INPs active above -7 °C are deactivated after a single freeze-thaw cycle, those active below -7 °C are typically unaffected even after three freezing cycles (Polen et al., 2016). Further, experiments

with INPs from soils show a remarkable stability of the ice nucleation temperature over tens of repeated melting and freezing cycles, with standard deviations of 0.2 °C (Vali, 2008). Furthermore, Wright et al. (2013) reported similar results for rain water samples. Since we analysed the collected crystals within minutes after melting, we can also exclude changes due to storage (i.e. aging), which has been observed with bulk snow samples over the course of days or weeks (Stopelli et al., 2014).

Surface frost can be a strong source of very small (i.e. < 100 µm), secondary ice crystals at Jungfraujoch (Lloyd et al., 2015) and at other mountain stations (Beck et al., 2018). During 7 of 10 sampling events air masses approached from northwest. The terrain falls off steeply in this direction and reaches the average observed cloud base (~1000 m below Jungfraujoch, Table 1) within a horizontal distance of about 2 km. At an average wind velocity of 8 m s$^{-1}$ from this direction the distance is covered within less than 5 minutes, which is too short for small, broken off frost crystals to grow to the average size of the crystals we have analysed (average of 3.1 mm). Even in most favourable conditions a dendrite would not grow to 1 mm diameter within that time (Takahashi et al., 1991). Therefore, it seems unlikely that dendrites which were not associated with an INP$_{-17}$ had grown from particles of hoar frost emitted by surfaces in the vicinity of Jungfraujoch.

The ice crystal habits encode information about the growth temperature of the crystals not their formation temperature. The growth temperature from -20 °C to -70 °C is the so-called "polycrystalline regime" dominated by crystal shapes with a range of different angles between branches or plates extending in three dimensions (Bailey and Hallett, 2009). These crystals will continue to grow when falling into warmer layers of air, as long as these layers are supersaturated with respect to ice. Otherwise, the crystals will sublimate. The growth habit of the falling crystals may change depending on temperature and supersaturation, but it will remain polycrystalline and irregular (c.f. Fig. 6 and 7 in Bailey and Hallett, 2009). Polycrystalline ice particles are highly unlikely to grow into the kind of crystals we have sampled, which had the same angle (60°) between all branches, and branches only extending in a single plane (i.e. dendrites; c.f. Schwarzenboeck et al., 2009). The lowest temperature at which the formation of the collected crystals may have been initiated is very likely above -20 °C because crystals formed by homogeneous freezing or INPs activated at temperatures below -20 °C would have resulted in polycrystalline crystals (Bailey and Hallett, 2009), a different habit than that of the crystals we had collected. Furthermore, according to Furukawa and Takahashi (1999) a dendrite falls about 400 m while growing to a diameter of around 3 mm. Given a diabatic lapse rate of 7.5 °C km$^{-1}$ an initial ice crystal may have been generated in 3 °C colder conditions than where its growth into a 3 mm dendrite was completed. However, as the deposition velocity of a tiny initial ice crystal is small, the initial ice formation will unlikely have occurred at much higher altitudes than where the main growth into dendrites occurred. Based on these findings, the information of growth temperature encoded in the habit of a crystal enables an assumption about the temperature range at which the crystal formed. For dendritic crystals, we can assume that the initial formation temperature is likely above -20 °C. Even if we consider all crystals which contained an INP active between -12 °C and -20 °C a large fraction of them (81%) remains to be considered the product of secondary ice formation.

The presence of INPs active at temperatures colder than -17 °C associated with the collected crystals might be explained by riming, i.e. the collection of cloud droplets containing such particles not activated as INP (i.e. scavenged INP) because ambient temperatures were not cold enough (Table 1). A majority of our crystals were rimed or densely rimed. The median concentration of INP$_{-25}$ in the rime samples collected on an impactor plate at Jungfraujoch was about 1100 ml$^{-1}$ during the period from 15 February to 12 March. Since 41% (background subtracted) of our crystals contained an INP$_{-25}$, the average mass of rime associated with a single crystal ($m$) must have been about 4.9 x 10$^{-4}$ g (see Eq. 1). This is about twice as much as the difference in mass ($\sim$2 x 10$^{-4}$ g) between rimed and un-rimed dendrites of 3 mm diameter found at Mount Tokachi, Hokkaido (Nakaya and Terada, 1935). The median of INPs active at -17 °C or warmer in rime was 16 ml$^{-1}$. Therefore, less than 1% of the crystals we have analysed might have scavenged an INP through riming that was active at -17 °C or warmer (16 [INP$_{-17}$ g$^{-1}$ rime] x 4.9 x 10$^{-4}$ [g rime crystal$^{-1}$]).

**Conclusion**

The habit of a planar, branched ice crystal, growing exclusively between -12 °C and -17 °C, enables the verification of whether it derived from primary or secondary ice formation based on a number of reasonable assumptions. Although the required experimental procedure including refreezing of dendrites using a drop freezing assay has a low throughput ($\sim$15 minutes for two ice crystals) it can provide an estimate for the ice multiplication factor around -15 °C, even when it is smaller than 10, unlike previous *in situ* approaches. The low throughput only provides for averaging over prolonged sampling periods and not for investigating single clouds. The factor we observed was much smaller than the 'several orders of magnitude' sometimes inferred from circumstantial evidence. Furthermore, we do not know whether the multiplication factor we found for dendrites is the same for other crystal habits found in the same MPCs. Because the estimated cloud base temperature was mostly below 0 °C during our observations, rime splintering and ice-ice collision breakup are more likely to have played a relevant role as secondary ice formation processes, as compared to droplet shattering (Sullivan et al., 2018). Whichever process was operating, it must have produced very small fragments, otherwise there would not have grown singular, regular, branched crystals (e.g. dendrites) from them. To learn more about the occurrence of secondary ice formation in moderately supercooled clouds, we think it would be valuable to repeat these experiments in other meteorological conditions or in other locations, such as those where most crystals were previously found to contain an insoluble particle in their centre or where they are less rimed. Less riming is likely to generate a smaller number of fragments by ice-ice collision breakup of dendrites as parametrized by Phillips et al. (2017). Under such conditions we would expect to find a smaller ice multiplication factor. This study analyses the refreezing ability of single sampled crystals and has shown that growth temperature information contained in the habit of an ice crystal can be a starting point to quantify ice multiplication in clouds.

**Data availability**

The data are available from the authors upon request.

**Competing interests**

The authors declare that they have no conflict of interest.

5   **Acknowledgments**

The authors would like to thank Sylvia C. Sullivan and the two anonymous referees for their valuable suggestions and comments during the review process which significantly improved this paper. We are grateful for the comments provided by Jann Schrod at Goethe University of Frankfurt on a draft of the paper. We also would like to thank the International Foundation High Altitude Research Stations Jungfraujoch and Gornergrat (HFSJG), 3012 Bern, Switzerland, for providing the 10   infrastructure and making it possible to work comfortably within mixed-phase clouds. Special thanks go to Joan and Martin Fischer, and Christine and Ruedi Käser, the custodians of the station, for their great support during the field campaign. Meteorological data at Jungfraujoch have been provided by MeteoSwiss, the Swiss Federal Office of Meteorology and Climatology. We are grateful to Maxime Hervo from MeteoSwiss for the provision of the ceilometer data collected at Kleine Scheidegg. We acknowledge the group of Ulrike Lohmann for borrowing their cloud droplet samplers and for fruitful 15   discussions. This study was financially supported by the Swiss National Science Foundation (SNF) through grant number 200021_169620. Participation of JMC in the campaign on Jungfraujoch was made possible through the SNF Scientific Exchanges Programme, grant number IZSEZ0_179151.

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

**Table 1.** Sampling periods including the date and the time span, numbers of analysed crystals (n), mean air temperature (T) (and standard deviation), mean wind velocity (u) (and standard deviation) and mean wind direction (dd) at Jungfraujoch; mean height of the station above cloud base (zB) and estimated mean cloud base temperature (CBT).

| Date | Time span | n | T | u | dd | zB | CBT |
|---|---|---|---|---|---|---|---|
| dd/mm/yyyy | UTC | - | °C | m/s | - | m | °C |
| 15/02/2018 | 07:30 - 21:50 | 38 | -7.0 (0.8) | 13.5 (2.1) | NW | 944 | 0.1 |
| 16/02/2018 | 09:30 - 16:30 | 29 | -8.7 (0.2) | 9.0 (2.4) | NW | 1239 | 0.6 |
| 17/02/2018 | 09:40 - 23:40 | 42 | -8.6 (1.7) | 5.8 (1.9) | NW | 693 | -3.3 |
| 23/02/2018 | 10:30 - 21:20 | 20 | -14.8 (0.6) | 11.9 (1.6) | SE | 365 | -12.1 |
| 06/03/2018 | 12:20 - 19:20 | 14 | -13.1 (0.1) | 5.5 (0.8) | NW | 1284 | -3.4 |
| 07/03/2018 | 08:00 - 16:40 | 23 | -15.7 (0.8) | 4.5 (2.6) | NW | 1001 | -8.2 |
| 10/03/2018 | 09:30 - 12:50 | 11 | -6.8 (0.3) | 5.1 (1.3) | E | 196 | -5.4 |
| 11/03/2018 | 15:40 - 17:00 | 6 | -9.8 (0.1) | 13.1 (1.4) | SE | 1485 | 1.3 |
| 12/03/2018 | 09:10 - 11:10 | 12 | -11.4 (0.1) | 6.2 (0.7) | NW | 878 | -4.8 |
| 22/03/2018 | 15:50 - 22:30 | 34 | -15.2 (1.2) | 12.4 (1.5) | NW | 1079 | -7.1 |

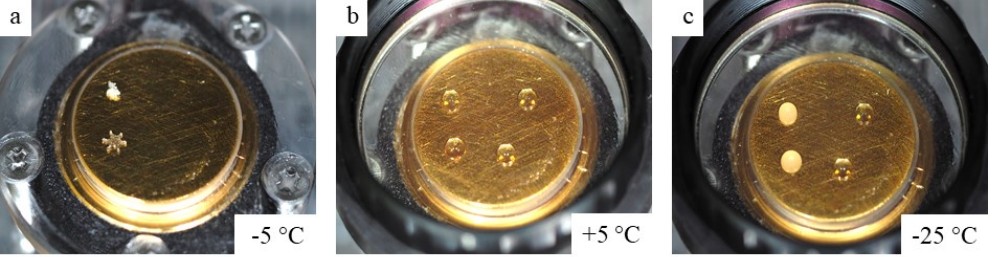

**Figure 1.** Illustration of one ice crystal droplet freezing experiment. Transparent droplets are liquid. **(a)** Two single crystals on the cold-stage (note: the cold-stage was set to below 0 °C for this image and the upper crystal is not a dendrite). **(b)** Melted ice crystals with addition of 3 µL ultrapure water to increase the detection volume (left) and two 3 µL control droplets of the same ultrapure water (right). **(c)** The frozen sample (left) and supercooled control (right) droplets after cooling to -25 °C.

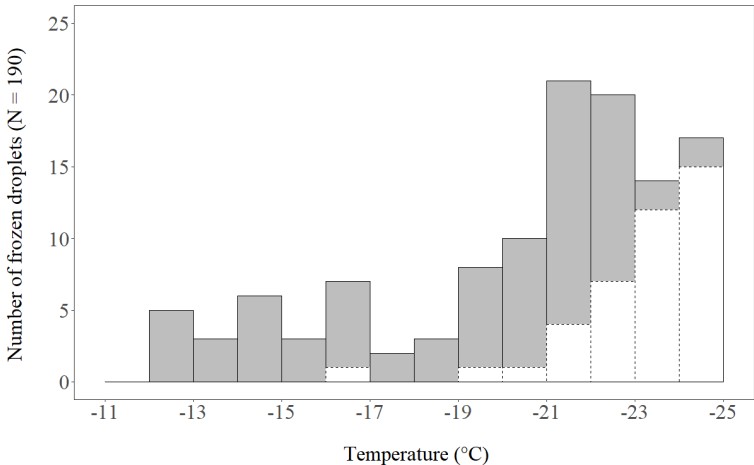

**Figure 2.** Number of planar, branched ice crystals that re-froze on a cold-stage after having been molten (grey bars with solid contour), thereby confirming they contained an INP active within the respective 1 °C temperature step. Of 190 crystals analysed, 24 re-froze at -17 °C or warmer (INP-17). The white bars with dashed contour indicate the number of frozen control droplets. The total number of control droplets was 190 as well.

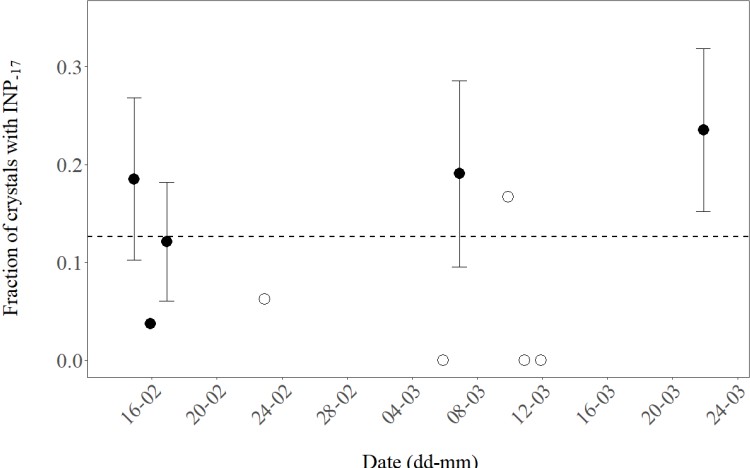

**Figure 3.** Daily fraction of ice crystals with INPs active at -17 °C or warmer (INP-17) observed for 10 days during February and March 2018. The number of crystals analysed per day was between 21 and 34 (closed symbols) or less (3 to 16, open symbols). Error bars indicate an estimate of the standard deviation (proportional to √INP-17) for days when at least four crystals with INP-17 were found. The dashed line shows the mean value of the pooled data (190 analysed crystals).