# Peer review of "New type of evidence for secondary ice formation at around -15 °C in mixed-phase clouds"

_Atmospheric Chemistry and Physics, 2018_

## Referee Comment (RC1) · Anonymous Referee #1 · 28 Aug 2018

**Direct evidence for secondary ice formation at around -15°C in mixed-phase clouds**

General comments

The manuscript shows results of cold stage tests from samples taken at Jungfraujoch with the aim of illustrating secondary ice formation at an "individual hydrometeor level". These analyses could yield quantitative estimates of ice crystal enhancement, but the data are too few to make a publication-worthy conclusion in my opinion. The authors note that Hoffer and Braham attempted a similar per-particle analysis more than 50 years ago. Their sample size was 300 snow pellets, 150% that presented here, and they note in their abstract that "a firm statement could not be made as the number of observations is limited." The burden is on the authors to explain why it is sufficient to show ground-based data from only 10 days.

Thereafter, the introduction needs to be expanded in my opinion. Right now, there is not a thorough discussion of existing literature. Approximate values and measurement techniques for INPs and IRs in mixed-phase clouds should be mentioned, in particular the abundance of measurements from Jungfraujoch (e.g. with the Ice Selective Inlet (Kupiszewski et al. 2015), the Ice Counterflow Virtual Impactor (Mertes et al. 2007), and the Horizontal Ice Nucleation Chamber (Lacher et al. 2017) as discussed in *Cziczo et al.* Measurements of Ice Nucleating Particles and Ice Residuals).

The analyses also need to be fleshed out. A more complete picture of the meteorology could be given by including the range and variability of air temperatures and wind velocities during the sampling periods. If photos of all the crystals were taken with a high-quality camera, some of these should be shown. Is there a more rigorous means of classifying the crystals than what is "considered to be planar, branched"? If the size of the crystals was measured with ImageJ, could some of these statistics also be presented? In Section 2.3, there is also a mention of rime analyses with a second cold plate, but it was not clear to me how this fit in. The results shown in Figures 2 and 3 are from pristine, unrimed dendrites, right?

With additional data and stronger analysis, more could be gleaned from this study. If the cold plate measurements are subject to any contamination, then 12.6% of the droplets refreezing is actually an overestimation. And a limited crystal geometry has been used to define secondary ice; at -15°C and lower supersaturations, other geometries are possible. So perhaps the multiplication factor of 8 is more of a lower bound. Quantitative estimates of this factor are needed for models, and field measurements at the hydrometeor level, rather than the bulk cloud level, are a new, if labor intensive, technique.

Specific comments

Page 1, Lines 18-20: The conclusion that "secondary ice can be observed at temperatures around -15°C" is not an especially compelling one, given that many previous studies have already shown this. Is there a hypothesized mechanism? Or was observed multiplication factor higher under certain conditions?

Page 1, Line 23 – "*These freezing pathways*" as there can be contact or deposition or immersion freezing.

Page 1, Line 26 – A few additional, more recent observations might be cited. For example, Lasher-Trapp et al. *JAS* [2016], Ladino et al. *GRL* [2017], and Jackson et al. *ACPD* [2018].

Page 2, Line 19 – For completeness, you could mention the correction of such shattering artifacts in more recent data by inter-arrival time algorithms and K-tip probes.

Page 2, Line 22 – I would define rime when you first discuss rime splintering above in Lines 4-5.

Section 2.2 and Figure 2 – The authors have taken a number of concerns about cold-stage measurements into consideration with their setup, which I appreciate. I would cite Tobo 2016 for the use of a semi-sold, hydrophobic substrate, and you might mention the possibility that INP settle out or aggregate within your large-volume droplet [e.g., Emerstic et al. 2015 *ACP*]. I am still concerned, however, that 20% of the control droplets have frozen by -25°C, almost 10°C above the threshold temperature for homogeneous freezing. Could the estimated enhancement factor be adjusted to account for these "false positives"?

Page 4, Line 1 – I would add a sentence that summarizes what this 'global classification scheme' is because it is not so widely used, as far as I know.

Page 4, Line 27 – Is there a reason that the "custom-built cold stage" used for single crystal analysis was not also used for the rime?

Page 4, Lines 28-29 – I am not sure what is meant by "droplets of molten rime". You are melting the aggregation of frozen droplets and then refreezing them upon a cold plate? Or somehow separating the droplets within a single aggregate? Please clarify here.

Page 5, Lines 19-21 – Measurement uncertainty and / or variability for this estimate needs to be included.

Page 5, Line 32-Page 6, Line 1 – The mention of INP from soils does not seem particularly relevant to me, as those will not be the INP source at Jungfraujoch.

Page 6, Lines 5-14 – Blowing snow is a very important consideration here, given several existing studies on this mechanism at Jungfraujoch. You are considering pristine dendrites here, right? Otherwise, there is the potential for riming growth, not just depositional growth.

Table 1 – For periods that last as much as 14 hours, it would be more rigorous to give mean and standard deviation for values like air temperature / wind velocity since a single value will not be characteristic. Are there are any vertical wind measurements?

---

## Short Comment (SC1) · 5 Sep 2018

We thank Anonymous Referee #1 for his or her assessment and valuable suggestions. Before replying point-by-point on behalf of all Co-Authors in an AC we would first like to clarify the main issue of concern in a personal Short Comment (SC):

RC: "These analyses could yield quantitative estimates of ice crystal enhancement, but the data are too few to make a publication-worthy conclusion in my opinion. The authors note that Hoffer and Braham attempted a similar per particle analysis more than 50 years ago. Their sample size was 300 snow pellets, 150% that presented here, and they [Hoffer and Braham, author's note], note in their abstract that "a firm statement could not be made as the number of observations is limited." The burden is

on the authors to explain why it is sufficient to show ground-based data from only 10 days."

SC: An ice multiplication factor can be estimated from the total number of analysed snow crystals divided by the number of snow crystals found to have formed through heterogeneous freezing. Hoffer and Braham had analysed 300 snow crystals (or pellets) but "...these pellets did not originate through the heterogeneous freezing of cloud drops." (Hoffer and Braham, 1962). Their data (300/0) therefore did not provide for an estimate of the ice multiplication factor in the clouds they had studied. Therefore, "a firm statement could not be made".

In our study we analysed a total of 190 crystals of which 24 had formed through heterogeneous freezing. Therefore, we can conclude that the ice multiplication factor in the clouds we have studied was around eight (190/24). The uncertainty associated with this factor is about 20% (square root of 24 divided by 24). Let us assume, we had done this kind of investigation in another location where one in two crystals is formed through heterogeneous freezing. In such a location we would have been able to determine the ice multiplication factor (two) with an uncertainty of 20% by analysing only 50 crystals in total.

To conclude, it is not the total number of snow crystals analysed but the number of snow crystals found to have formed through heterogeneous freezing that determines how firm a conclusion can be drawn.

Claudia Mignani and Franz Conen

---

## Referee Comment (RC2) · Anonymous Referee #1 · 10 Sep 2018

Thanks very much for your response concerning the uncertainty in the ice multiplication factor estimate given your sample. My concern is more related to the uncertainty in the representativeness of your sample for the population. Assuming that "the population" here is ice in a mixed-phase cloud, then you can estimate the population size .. say conservatively that the cloud is 2 km deep and has an equivalent radius of 3 km, then it already has a volume on the order of 10^10 cubic meters. Even if the ice crystal concentration in the cloud is only a crystal per cubic meter, you have sampled a very small portion of the population for which you are making a conclusion. This is how I am thinking, but I understand that there are all sorts of subtleties related to representativeness and that your collection process is laborious, so let us see what other reviewers say.

---

## Referee Comment (RC3) · Anonymous Referee #2 · 11 Sep 2018

Authors present the experimental work where they collected the snow crystals, melted the crystals and visually observed the freezing of the crystal droplet. These results were used to understand more about the secondary ice formation and ice multiplication factors. These questions are challenging, and the community needs an understanding of these cloud processes for better representation in the cloud model. However, this study lacks appropriate experimental technique/methodology to answer these questions, and for this reason, the paper is not ready for the publication. I'm not sure if the major review could improve the paper further as substantial experimental work is involved.

There are a number of issues in the present experimental study. If no INP was observed within the crystal, it does not mean that crystal was formed through secondary

ice formation mechanism. It is possible that a INP may have induced nucleation of ice, and still while INP is floating within the atmosphere may have detached from the ice crystal because the crystal evaporated or through some turbulent process. Now, this crystal when sampled had no INP. It is also possible that INP is present, but was deactivated while it went transformation (change in physical and chemical properties) during sampling, heating or droplet preparation. There are numerous studies in the literature that discusses the deactivation of INP. Such discussion is missing. Experiments are needed that investigate the ice nucleation efficiency of crystal melted droplets up to -37 degC (below this temperature homogeneous freezing is the dominant mode of ice nucleation) to understand more about the insoluble INPs, but for soluble INPs experiments should be investigated at homogeneous freezing temperatures too. Without such results, the conclusions regarding secondary ice formation cannot be inferred. Supporting experiments are needed to say why there was no INPs present (page 5 line 14). It would be just that the limitation of the experimental setup. In this study, the sample collection onto the cold stage is not done in clean air conditions. It is possible that crystals were contaminated with room air particles. Further, it is possible that these particles may have induced nucleation of ice but not the primary INP (the first INP that was responsible for freezing the droplet in the atmosphere before sampling). Without knowing the composition of residue it is difficult to infer which INP (primary or room air particulates) was responsible for freezing. It is not clear how section 2.3 supports the secondary ice formation analysis. Details such as validation and performance calibration of the cold stage (shown in Fig 1) under different temperature and humidity conditions are missing. Any results from previous studies who had attempted to study secondary ice formation should be shown in Figure 2 and 3. Discussion regarding nature of INP is missing. What are their composition and size?

One should use Ice-CVI (Mertes et al 2007) to sample only ice crystals, sublimate/evaporate these crystals, count the residues and investigate the ice nucleation propensity of a single residue. By comparing inlet ice crystal and residue concentrations one can infer some understanding regarding secondary ice formation.

S. Mertes , B. Verheggen , S. Walter , P. Connolly , M. Ebert , J. Schneider , K. N. Bower , J. Cozic , S. Weinbruch , U. Baltensperger & E. Weingartner (2007) Counter-flow Virtual Impactor Based Collection of Small Ice Particles in Mixed-Phase Clouds for the Physico-Chemical Characterization of Tropospheric Ice Nuclei: Sampler Description and First Case Study, Aerosol Science and Technology, 41:9, 848-864, DOI: 10.1080/0278682070150188.

---

## Author Comment (AC1) · 28 Sep 2018

**Authors' response to Anonymous Referee #1**
**Review received and published: 28 August 2018**

> For clarity and easy visualization, the referee's comment is copied here in black. The authors' replies are in blue font with an increased indent below each of the referee's statements. Page and line numbers refer to online ACPD version.

General comments

The manuscript shows results of cold stage tests from samples taken at Jungfraujoch with the aim of illustrating secondary ice formation at an "individual hydrometeor level". These analyses could yield quantitative estimates of ice crystal enhancement, but the data are too few to make a publication-worthy conclusion in my opinion. The authors note that Hoffer and Braham attempted a similar per-particle analysis more than 50 years ago. Their sample size was 300 snow pellets, 150% that presented here, and they note in their abstract that "a firm statement could not be made as the number of observations is limited." The burden is on the authors to explain why it is sufficient to show ground-based data from only 10 days.

> We thank Anonymous Referee #1 for his/her assessment and valuable suggestions. In a 'short comment' we have already clarified the valid point about the sample size. We will discuss that the number of snow crystals found to have formed through heterogeneous freezing determines how firm a conclusion can be drawn and not the total number of snow crystals analysed and we will add the uncertainty of the multiplication factor during revisions. The uncertainty associated with the observed multiplication factor is about 20% (square root of 24 divided by 24). The detailed answer can be found in the short comment posted on 5th September 2018.

Thereafter, the introduction needs to be expanded in my opinion. Right now, there is not a thorough discussion of existing literature. Approximate values and measurement techniques for INPs and IRs in mixed-phase clouds should be mentioned, in particular the abundance of measurements from Jungfraujoch (e.g. with the Ice Selective Inlet (Kupiszewski et al. 2015), the Ice Counterflow Virtual Impactor (Mertes et al. 2007), and the Horizontal Ice Nucleation Chamber (Lacher et al. 2017) as discussed in Cziczo et al. Measurements of Ice Nucleating Particles and Ice Residuals).

> We will expand our introduction and refer to the mentioned studies to make clear already in the introduction the novelty of our approach. By sampling small ice particles (few tens of micron in aerodynamic diameter) at Jungfraujoch earlier studies were not able to separate ice which had formed in clouds from aerosolised parts of hoar frost growing on surrounding surfaces (Lloyd et al., 2015; Farrington et al., 2016; Beck et al., 2018). By sampling larger, regular ice particles (e.g. dendrites) we minimised the influence of local surfaces on our results (page 6 line 10-14) and can draw a conclusion regarding secondary ice formation within mixed-phase clouds.

The analyses also need to be fleshed out. A more complete picture of the meteorology could be given by including the range and variability of air temperatures and wind velocities during the sampling periods. If photos of all the crystals were taken with a high-quality camera, some of these should be shown. Is there a more rigorous means of classifying the crystals than what is "considered to be planar, branched"? If the size of the crystals was measured with ImageJ, could some of these statistics also be presented? In Section 2.3, there is also a mention of rime analyses with a second cold plate, but it was not clear to me how this fit in. The results shown in Figures 2 and 3 are from pristine, unrimed dendrites, right?

We will provide the range and variability of air temperatures and wind velocities in a revised version. We are ready to show representative pictures of the crystals that were taken in the main paper or all pictures as supporting information. As mentioned on page 3 line 31, we classified the crystals by habit and riming degree using the 'global classification scheme'. There are some variations in crystal shape. We considered as planar, branched crystals, crystals that can be classified into the following classes: R1c, R2c, R3a, R4a, P4, P3, P2 according to Kikuchi et al. (2013) (see first paragraph in Section 3; please note that the paper by Kikuchi et al. contains representative images of all the crystal classes mentioned in our paper). Figures 2 and 3 show the results of the 190 crystals. A large fraction of them are 'rimed dendrites' or 'densely rimed dendrites' (see page 5 line 6). Rime itself was analysed to determine the fraction of analysed crystals that possibly had scavenged through riming an INP active at -17 °C or warmer. This fraction was smaller than 1% (page 6, lines 32-33)

With additional data and stronger analysis, more could be gleaned from this study. If the cold plate measurements are subject to any contamination, then 12.6% of the droplets refreezing is actually an overestimation. And a limited crystal geometry has been used to define secondary ice; at -15°C and lower supersaturations, other geometries are possible. So perhaps the multiplication factor of 8 is more of a lower bound. Quantitative estimates of this factor are needed for models, and field measurements at the hydrometeor level, rather than the bulk cloud level, are a new, if labor intensive, technique.

With the same number of control droplets as droplets from crystals we assessed potential contamination. For temperatures at which the analysed crystals had formed (-12 °C to -17 °C) only 0.5% (1 in 190) of the droplets were contaminated. Indeed, at lower supersaturations other crystal geometries are possible at around -15 °C. However, as we were sampling within mixed-phase clouds, we were always within highly supersaturated conditions. We would like to recall that our aim was to find reliable evidence for secondary ice formation at around -15 °C in clouds. For this reason, we had to exclude as rigorously as possible the influence of secondary ice formed and aerosolised from local surfaces (e.g. hoar frost). This requirement called for selecting crystals with a regular shape that forms in clouds and a size large enough to tell they have not grown from splinters emitted locally (see page 6, lines 5-14). We agree that the estimated ice multiplication factor may therefore be a lower bound, a point we will make clear when revising the manuscript.

Specific comments

Page 1, Lines 18-20: The conclusion that "secondary ice can be observed at temperatures around -15°C" is not an especially compelling one, given that many previous studies have already shown this. Is there a hypothesized mechanism? Or was observed multiplication factor higher under certain conditions?

As far as we know, no previous study has directly observed secondary-produced ice at around -15 °C in natural mixed-phase clouds. What has been reported were large discrepancies between number concentrations of ice crystals and INPs. We could only speculate which mechanisms is responsible for the secondarily produced ice by referring to the papers by Field et al. (2017) and Sullivan et al. (2018), both cited in the manuscript. As shown in Figure 3 the daily fraction of primary crystals was relatively constant and varied
around the mean value of the pooled data. When considering the uncertainty of those days
where we had at least four primary crystals (black dots), their means are not distinguishable
from the pooled data (mean +/- standard deviation of the pooled data).

Page 1, Line 23 – "These freezing pathways" as there can be contact or deposition or immersion
freezing.

Correct, thank you.

Page 1, Line 26 – A few additional, more recent observations might be cited. For example, Lasher-
Trapp et al. JAS [2016], Ladino et al. GRL [2017], and Jackson et al. ACPD [2018].

We will add them.

Page 2, Line 19 – For completeness, you could mention the correction of such shattering artifacts in
more recent data by inter-arrival time algorithms and K-tip probes.

Thank you for mentioning; we will do that.

Page 2, Line 22 – I would define rime when you first discuss rime splintering above in Lines 4-5.

O.k., we will define rime there.

Section 2.2 and Figure 2 – The authors have taken a number of concerns about cold-stage
measurements into consideration with their setup, which I appreciate. I would cite Tobo 2016 for
the use of a semi-sold, hydrophobic substrate, and you might mention the possibility that INP settle
out or aggregate within your large-volume droplet [e.g., Emerstic et al. 2015 ACP]. I am still
concerned, however, that 20% of the control droplets have frozen by -25°C, almost 10°C above the
threshold temperature for homogeneous freezing. Could the estimated enhancement factor be
adjusted to account for these "false positives"?

The estimated enhancement factor relates to the temperature window in which the
collected ice crystals were likely to have formed (-12 °C to -17 °C). In this temperature
window we had only one false positive in 190 tested controls. The number of droplets frozen
by -25 °C only plays a role when estimating the average mass of rime associated with a single
crystal. In this estimate we have accounted for the false positives (subtracted frozen controls
from frozen droplets; please see page 6, line 27).

Page 4, Line 1 – I would add a sentence that summarizes what this 'global classification scheme' is
because it is not so widely used, as far as I know.

We will add a sentence that summarizes the global classification scheme.

Page 4, Line 27 – Is there a reason that the "custom-built cold stage" used for single crystal analysis
was not also used for the rime?

Unlike traditional cold plate systems, the custom-built cold stage is mainly made to be easily
field transportable to remote locations. It was however not used for the rime samples as it
has a rather small surface (surface diameter of 18 mm, page 4 line 5). Analysing rime with it
would have led to less measurement time for the single crystals. Our goal was to get as
much measurement time for the single crystals as possible. This requires a cold stage which
is ready to be used when the specific type of crystals precipitate. The second cold stage i.e.
the NOAA Drop Freezing Cold Plate has a larger surface, therefore more droplets can be placed on it, which is convenient for the rime analysis. Please note that the NOAA Drop Freezing Cold Plate requires an external refrigerated circulation bath, lined power and is relatively large. We could not put it into the anteroom and analyse single crystals. We used the most suited cold stage type available for each sample type. We will add the reason why we used two different cold plate systems in the revised manuscript and discuss whether the results of both plates are comparable

Page 4, Lines 28-29 – I am not sure what is meant by "droplets of molten rime". You are melting the aggregation of frozen droplets and then refreezing them upon a cold plate? Or somehow separating the droplets within a single aggregate? Please clarify here.

Indeed, we are melting the aggregation of the frozen rime droplets and then refreezing them. We will clarify this in a revised manuscript.

Page 5, Lines 19-21 – Measurement uncertainty and / or variability for this estimate needs to be included.

Measurement uncertainty will be included on page 5, lines 19-21.

Page 5, Line 32-Page 6, Line 1 – The mention of INP from soils does not seem particularly relevant to me, as those will not be the INP source at Jungfraujoch.

This sentence presents one of three examples from the literature that illustrate the ice nucleation temperature stability during repeated melting and freezing and therefore we think that it is worthwhile to mention it. Besides that, we think that aerosolised soil particles, or soil dust, potentially emitted from fields in northern Italy, southern France, southern Germany, and the Swiss Plateau might make a relevant contribution to INPs active at relatively high temperatures (i.e. > -17 °C) at Jungfraujoch. Note that the most prominent particle classes (reflecting particles in the size range between 0.5 um and 5 um) determined at Jungfraujoch were carbonaceous particles (Hinz et al., 2005). Furthermore, most of the fields within the fetch of Jungfraujoch are not covered by snow during winter and wind blown dust emissions are relatively high during that season in Europe (Korcz et al., 2009).

Page 6, Lines 5-14 – Blowing snow is a very important consideration here, given several existing studies on this mechanism at Jungfraujoch. You are considering pristine dendrites here, right? Otherwise, there is the potential for riming growth, not just depositional growth.

We are considering planar, branched crystals including rimed crystals. This is the reason why we have also analysed the INP spectra of rime itself. Our results show that less than 1% of the analysed crystals may have scavenged an INP active at a similar temperature as the INP which might have catalysed the formation of a dendrite (page 6 lines 23-34).

Table 1 – For periods that last as much as 14 hours, it would be more rigorous to give mean and standard deviation for values like air temperature / wind velocity since a single value will not be characteristic. Are there are any vertical wind measurements?

O.k., we will add standard deviations for air temperature and wind velocity. No vertical wind measurements were taken though.

**Authors' response to interactive comment from Anonymous Referee #1**
**Interactive comment received and published: 10 September**
Thanks very much for your response concerning the uncertainty in the ice multiplication factor
estimate given your sample. My concern is more related to the uncertainty in the representativeness
of your sample for the population. Assuming that "the population" here is ice in a mixed-phase
cloud, then you can estimate the population size ..say conservatively that the cloud is 2 km deep and
has an equivalent radius of 3 km, then it already has a volume on the order of $10^{10}$ cubic meters.
Even if the ice crystal concentration in the cloud is only a crystal per cubic meter, you have sampled
a very small portion of the population for which you are making a conclusion. This is how I am
thinking, but I understand that there are all sorts of subtleties related to representativeness and that
your collection process is laborious, so let us see what other reviewers say.

The number of crystals we have sampled and analysed is indeed a very small fraction of all
crystals in all clouds that have passed Jungfraujoch between 15 February and 22 March
2018. If we had sampled the crystals from a small fraction of a cloud volume and would
extrapolate our findings to a much larger volume in which primary and secondary crystals
are very heterogeneously distributed, we would face a problem. However, we have sampled
crystals on 10 days in different clouds spread over a period of 36 calendar days. To estimate
the total path of clouds crossing Jungfraujoch during our sampling events, we can multiply
the sampling duration of an event with the mean wind speed (values in Table 1). By doing so
and taking the sum of all the sampling events, we get a total path of 2368 km within clouds
along which we have taken our samples. We think that this is a representative distance of
cloud passage and thus a representative sample for this year's winter clouds at Jungfraujoch.
Figure 2 shows that the ice multiplication factor for individual days is similar to the mean of
the pooled data, considering the larger uncertainty of daily estimates as compared to the
estimate for the whole period. Hence, the average of the period is not subject to bias of a
single day with substantially different multiplication factor from the rest of the days.

**References**

Beck, A., Henneberger, J., Fugal, J. P., David, R. O., Lacher, L., and Lohmann, U.: Impact of
surface and near-surface processes on ice crystal concentrations measured at mountain-top
research stations, Atmos. Cham. Phys., 18, 8909-8927, 2018.

Farrington, R. J., Connolly, P. J., Lloyd, G., Bower, K. N., Flynn, M. J., Gallagher, M. W., Field,
P. R., Dearden, C., and Choularton, T. W.: Comparing model and measured ice crystal
concentrations in orographic clouds during the INUPIAQ campaign, Atmos. Chem. Phys., 16,
4945-4966, doi:10.5194/acp-16-4945-2016, 2016.

Field, R. P., Lawson, R. P., Brown, P. R. A., Lloyd, G., Westbrook, C., Moisseev, D.,
Miltenberger, A., Nenes, A., Blyth, A., Choularton, D., Connolly, P., Buehl, J., Crosier, J., Cui,
Z., Dearden, C., DeMott, P., Flossmann, A., Heymsfield, A., Huang, Y., Kalesse, H., Kanji, Z. A.,
Korolev, A., Kirchgaessner, A., Lasher-Trapp, S., Leisner, T., McFarquhar, G., Phillips, V., Stith,
J., and Sullivan, S.: Secondary ice production: Current state of the science and
recommendations for the future, Meteorol. Monogr., 58, 7.1-7.20,
doi:10.1175/AMSMONOGRAPHS-D-16-0014.1, 2017.

Hinz, K.-P., Trimborn, A., Weingartner, E., Henning, S., Baltensperger, U., Spengler, B.:
Aerosol single particle composition at the Jungfraujoch, J. Aerosol Sci., 36, 123–145,
https://doi.org/10.1016/j.jaerosci.2004.08.001, 2005.

Korcz, M., Fudala, J., Klis, C.: Estimation of wind blown dust emissions in Europe and its
vicinity. Atmos. Environ., 43, 1410-1420, doi: 10.1016/j.atmosenv.2008.05.027, 2009.

Lloyd, G., Choularton, T. W., Bower, K. N., Gallagher, M. W., Connolly, P. J., Flynn, M.,
Farrington, R., Crosier, J., Schlenczek, O., Fugal, J., and Henneberger, J.: The origins of ice
crystals measured in mixed-phase clouds at the high-alpine site Jungfraujoch, Atmos. Chem.
Phys., 15, 12 953–12 969, doi:10.5194/acp-15-12953-2015, 2015.

Sullivan, S. C., Hoose, C., Kiselev, A., Leisner, T., and Nenes, A.: Initiation of secondary ice
production in clouds, Atmos. Chem. Phys., 18, 1593-1610, https://doi.org/10.5194/acp-18-
1593-2018, 2018.

---

## Author Comment (AC2) · 28 Sep 2018

**Authors' response to Anonymous Referee #2**
**Review received and published: 11 September 2018**
For clarity and easy visualization, the referee's comment is copied here in black. We have
divided his/her text into numbered sections to facilitate discussion. The authors' replies are
in blue font with an increased indent below each of the referee's statements. Page and line
numbers refer to online ACPD version.
Authors present the experimental work where they collected the snow crystals, melted the crystals
and visually observed the freezing of the crystal droplet. These results were used to understand
more about the secondary ice formation and ice multiplication factors. These questions are
challenging, and the community needs an understanding of these cloud processes for better
representation in the cloud model. However, this study lacks appropriate experimental
technique/methodology to answer these questions, and for this reason, the paper is not ready for
the publication. I'm not sure if the major review could improve the paper further as substantial
experimental work is involved. There are a number of issues in the present experimental study.
We thank Anonymous Referee #2 for openly sharing his/her ideas on our recent manuscript.
We agree that the questions about secondary ice formation and multiplication factors are
challenging and that the community needs to answer them to improve cloud
parametrizations in models. There are different approaches to answer these questions. The
work presented here addresses them by applying an unconventional, new method. The
study combines the growth temperature encoded in the habit of snow crystals with a drop
freezing assay and thus complements previous observations of secondary ice formation. Our
experimental technique is appropriate for detecting insoluble ice nucleation particles (INPs)
in single crystals and enables us to estimate with an uncertainty of about 20% the lower
bound of the ice multiplication factor in clouds during our sampling campaign at
Jungfraujoch. Herewith, we would like to dispel the referee's doubts and elucidate how we
will make use of the referee's comments in a revised version of the manuscript.
Furthermore, we are confident that our manuscript constitutes a valuable contribution to
ACP and we appreciate the opportunity to openly stand up for and constructively discuss our
work.
**Section1**
If no INP was observed within the crystal, it does not mean that crystal was formed through
secondary ice formation mechanism. It is possible that a INP may have induced nucleation of ice,
and still while INP is floating within the atmosphere may have detached from the ice crystal because
the crystal evaporated or through some turbulent process. Now, this crystal when sampled had no
INP.
We are not aware of any literature describing the mechanisms to which this statement could
refer to. Does Anonymous Referee #2 have supportive evidence for ice crystals losing their INP
through evaporation, sublimation, or through "some turbulent process" in the atmosphere
that are resulting in ice particles without INP?
In the atmosphere, ice nucleation has been observed at temperatures warmer than that of
homogeneous freezing (Ansmann et al., 2005). Four main pathways of heterogeneous freezing
have been identified: contact, deposition, condensation, and immersion freezing (Pruppacher
and Klett, 1997). In our study, we investigated freezing through the immersion freezing
mechanism. Immersion freezing refers to the initiation of ice nucleation by a solid and
insoluble INP immersed in a water droplet. To our understanding, the immersed INP will catalyse an initial crystal, in which the INP is embedded. This initial crystal then grows through vapour deposition. In this process, the INP in the initial crystal will increasingly become encased in ice that grows thicker around it. If this crystal then begins to sublimate, the ice covering the initial droplet surrounding the INP will become thinner again, which we expect to evolve rather uniformly from the outside (i.e. edge of the crystal) towards the inside (i.e. initial droplet that froze by immersion). The INP will be released from the ice only once the ice of the very initial frozen droplet has sublimated, resulting in an INP without ice, but not in ice without an INP.

Besides that, we are also not aware of observations that show how "some turbulent processes" may detach the INP from a crystal. How should the INP get out of the crystal structure? Is there at all relevant turbulent friction at the submillimetre-scale in the free atmosphere? We would however be happy to discuss such mechanisms in our manuscript if they have a theoretical or observational basis.

**Section 2**

It is also possible that INP is present, but was deactivated while it went transformation (change in physical and chemical properties) during sampling, heating or droplet preparation. There are numerous studies in the literature that discusses the deactivation of INP. Such discussion is missing.

Indeed, studies exist that discuss the deactivation of INP during transformation. In our experiment, the crystals were sampled below melting temperatures, and melted or "heated" to between +1 °C and +5 °C (page 4, line 12) before being analysed within the next minutes. It is not unusual to store INPs in water at +4 °C for several hours before analysis (e.g. Wilson et al., 2015). Studies reporting deactivation through heating typically refer to heating temperatures close to the boiling point of water (e.g. Christner et al., 2008).

There is also convincing evidence in the literature that INPs, which are active at temperatures relevant for our study, can be repeatedly activated, going through multiple cycles of freezing and melting. We have discussed and referred to these studies on page 5 line 30 to page 6 line 3. Furthermore, we have clearly formulated that our findings are based on the assumption that the cited evidence also applies to our samples, see page 5 line 23.

Several laboratory studies have investigated the role of coating of mineral dust particles and the related changes in ice nucleation efficiency (e.g. Knopf and Koop 2006, Cziczo et al., 2009; Kanji et al., 2018). Soluble coating or soluble INPs could be altered through melting or droplet preparation. However, the work presented here is not investigating the effect of soluble coating and neither of soluble INPs. Soluble INPs probably do not play a role at temperatures warmer than about -27 °C (Knopf et al., 2018, see their Fig. 5). Based on the referee's comment, we will emphasize in a revised version, that we are focusing on insoluble INPs in dendrites that can be activated through immersion freezing at temperatures above -17 °C for at least two freezing cycles (one when forming the crystal and one when doing the measurement).

**Section 3**

Experiments are needed that investigate the ice nucleation efficiency of crystal melted droplets up to -37 degC (below this temperature homogeneous freezing is the dominant mode of ice nucleation) to understand more about the insoluble INPs, but for soluble INPs experiments should be investigated at homogeneous freezing temperatures too. Without such results, the conclusions regarding secondary ice formation cannot be inferred.

Heterogeneous freezing at temperatures below -25 °C and homogenous freezing at even colder temperatures are certainly important topics of research, especially when investigating cold mixed-phase clouds or cirrus clouds. Observations have shown that an overwhelming majority of ice particles originate from supercooled liquid clouds at temperatures > -27 °C, which strongly suggests that the initial process of ice formation in mixed-phase clouds > -27 °C occurs through immersion freezing (Westbrook and Illingworth, 2011). Therefore, we assume that homogeneous freezing does not play an important role in mixed-phase clouds surrounding Jungfraujoch during our campaign where temperatures were clearly higher (see Table 1). Further, every experimental study has a limited parameter space. We set the frame for our study in the second part of the introduction. Briefly, our objective was to detect the presence of INPs active at around -15 °C in dendrites, which typically grow around that temperature. By investigating ice nucleation down to -25 °C we already expanded our measurements well beyond the necessary to answer the question to what proportion dendrites are the result of primary ice formation.

**Section 4**

Supporting experiments are needed to say why there was no INPs present (page 5 line 14). It would be just that the limitation of the experimental setup. In this study, the sample collection onto the cold stage is not done in clean air conditions. It is possible that crystals were contaminated with room air particles. Further, it is possible that these particles may have induced nucleation of ice but not the primary INP (the first INP that was responsible for freezing the droplet in the atmosphere before sampling). Without knowing the composition of residue it is difficult to infer which INP (primary or room air particulates) was responsible for freezing.

Indeed, it is very important in a first step to avoid contamination as much as possible and in a second step to quantify it. We examined contamination with control droplets of molecular grade water (blanks). If contamination, including deposition of INP from the room air would have been a problem, we would have seen it in the freezing of control droplets. As shown in Fig. 2 and discussed in the text, of 190 control droplets only one froze within the temperature range where the analysed crystals may have formed (-12 °C to -17 °C). Deposition of "room air-INPs" is only one out of several possible reasons why this control droplet may have frozen. Another reason could have been surface contamination of the cold stage. Please note that the control droplets were exposed to the same room air during the same time as were our sample. Thus, even without knowing the composition of residue, we can show, with the results of the control droplets, that INPs deposited from room can not have been responsible for the freezing of the crystal droplets.

It is not a limitation of our experimental setup that no INP active around -15 °C was found in a large proportion of the analysed dendrites. A possible explanation for the absence of INPs are crystals formed through secondary ice formation processes. Our results are consistent with findings and conclusions from other studies (page 1 line 26). Several studies measured much lower INP concentrations than ice crystal number concentrations in clouds by using different approaches and measurement techniques from ours.

**Section 5**

It is not clear how section 2.3 supports the secondary ice formation analysis. Details such as validation and performance calibration of the cold stage (shown in Fig 1) under different temperature and humidity conditions are missing.

The majority of analysed crystals were rimed. Rime could have added INPs active at around -15 °C to initial crystals (page 2 line 23-24). Therefore, we analysed not only (rimed) crystals but also rime itself (method in section 2.3). Our results show that riming had only a very minor influence on our results (page 6 line 23-34).

The cold stage was used to test for INPs in immersion freezing mode. Details of the cold stage as well as calibration can be found in the supplement, including the result of tests at a range of temperatures. We are not sure why we should perform validation and calibration at different humidity conditions. These would play a role only, if we would study deposition or condensation freezing.

**Section 6**

Any results from previous studies who had attempted to study secondary ice formation should be shown in Figure 2 and 3.

It would make sense to compare our results with previous studies. However, the results of previous studies are based on completely different approaches. Their results are not directly comparable to ours. One of the main differences is that we have analysed relatively large snow crystals (several millimetres in diameter) to make sure our results are not influenced by local surface sources of secondary ice formation. We will discuss differences regarding results and methodology between previous studies and this study in more detail in a revised version of our manuscript.

**Section 7**

Discussion regarding nature of INP is missing. What are their composition and size? One should use Ice-CVI (Mertes et al 2007) to sample only ice crystals, sublimate/evaporate these crystals, count the residues and investigate the ice nucleation propensity of a single residue. By comparing inlet ice crystal and residue concentrations one can infer some understanding regarding secondary ice formation.

Mertes et al. (2007) sampled very small ice particles, between 5 and 20 micron (aerodynamic diameter). Lloyd et al. (2015) concluded for Jungfraujoch that "hoar frost crystals generated at the cloud enveloped snow surface could be the most important source of cloud ice concentrations." The same may apply to other mountain stations (Beck et al., 2018). Therefore, repeating the experiments of Mertes et al. (2007) would tell us mainly about ice residues in hoar frost particles generated by local surfaces. This is not what we are interested in. We would like to know more about secondary ice formation in mixed-phase clouds themselves. This is the reason why we have sampled larger crystals with a regular shape that are unlikely to have resulted from surface processes and tested these crystals for the presence of INPs active within the temperature range they typically form.

**References**

Ansmann, A., Mattis, I., Müller, D., Wandinger, U., Radlach, M., Althausen, D., and Damoah, R.: Ice formation in Saharan dust over central Europe observed with temperature/humidity/aerosol Raman lidar, J. Geophys. Res. Atmos., 110, D18S12, doi:10.1029/2004JD005000, 2005.

Beck, A., Henneberger, J., Fugal, J. P., David, R. O., Lacher, L., Lohmann, U.: Impact of surface and near-surface processes on ice crystal concentrations measured at mountain-top research stations. Atmos. Chem. Phys. 18, 8909-8927, 2018 :

Christner, B. C., Morris, C. E., Foreman, C. M., Cai, R., Sands, D. C.: Ubiquity of biological ice nucleators in snowfall. Science, 319, 1214, doi: 10.1126/science.1149757, 2008.

Cziczo, D. J., Froyd, K. D., Gallavardin, S. J., Moehler, O., Benz, S., Saathoff, H., and Murphy, D. M.: Deactivation of ice nuclei due to atmospherically relevant surface coatings, Environ. Res. Lett, 4, 044 013, 2009.

Kanji, Z. A., Sullivan, R. C., Niemand, M., DeMott, P. J., Prenni, A. J., Chou, C., Saathoff, H., and Möhler, O.: Heterogeneous Ice Nucleation Properties of Natural Desert Dust Particles Coated with a Surrogate of Secondary Organic Aerosol, Atmos. Chem. Phys. Discuss https://doi.org/10.5194/acp-2018-905, in review, 2018.

Knopf, D. A., Alpert, P. A., Wang, B.: The role of organic aerosol in atmospheric ice nucleation: a review, ACS Earth Space Chem., 2, 168-202, doi:10.1021/acsearthspacechem.7b00120, 2018.

Knopf, D. A. and Koop, T.: Heterogeneous nucleation of ice on surrogates of mineral dust, J. Geophys. Res. Atmos., 111, D12 201, doi:10.1029/2005JD006894, 2006.

Lloyd, G., Choularton, T. W., Bower, K. N., Gallagher, M. W., Connolly, P. J., Flynn, M., Farrington, R., Crosier, J., Schlenczek, O., Fugal, J., and Henneberger, J.: The origins of ice crystals measured in mixed-phase clouds at the high-alpine site Jungfraujoch, Atmos. Chem. Phys., 15, 12 953–12 969, doi:10.5194/acp-15-12953-2015, 2015.

Mertes, S., Verheggen, B., Walter, S., Connolly, P., Ebert, M., Schneider, J., Bower, K. N., Cozic, J., Weinbruch, S., Baltensperger, U. and Weingartner E.: Counterflow Virtual Impactor Based Collection of Small Ice Particles in Mixed-Phase Clouds for the Physico-Chemical Characterization of Tropospheric Ice Nuclei: Sampler Description and First Case Study, Aeros. Sci. Tech., 41, 9, 848-864, DOI: 10.1080/02786820701501881, 2007.

Pruppacher, H. R. and Klett, J. D.: Microphysics of clouds and precipitation. 2nd edition, Kluwer Academic Publishers New York, Boston, Dordrecht, London, Moscow, 1997.

Westbrook, C. D., and Illingworth, A. J.: Evidence that ice forms primarily in supercooled liquid clouds at temperatures > −27 °C, Geophys. Res. Lett., 38, L14808, doi:10.1029/2011GL048021, 2011.

Wilson, T. W., Ladino, L. A., Alpert, P. A., Breckels, M. N., Brooks, I. M., Browse, J., Burrows, S. M., Carslaw, K. S., Huffman, J. A., Judd, C., Kilthau, W. P., Mason, R. H., McFiggans, G., Miller, L. A., Najera, J. J., Polishchuk, E., Rae, S., Schiller, C. L., Si, M., Temprado, J. V., Whale, T. F., Wong, J. P. S., Wurl, O., Yakobi-Hancock, J. D., Abbatt, J. P. D., Aller, J. Y., Bertram, A. K., Knopf, D. A., Murray, B. J.: A marine biogenic source of atmospheric ice-nucleating particles. Nature, 525, 234-238, doi:10.1038/nature14986, 2015.

---

## Referee Comment (RC4) · Anonymous Referee #2 · 3 Oct 2018

Thanks for providing more information about these experiments. However, authors do not address the concerns that are outlined. I will describe one example here. One of the conclusions of this study (page 5 main paper) is that if no INP was found in a crystal – this crystal was categorized as formed through the process of secondary ice formation. This is based on an observation that this particular crystal (now supercooled droplet) did not freeze until -25C. However, it is possible that this droplet may freeze at colder temperatures than -25C, and if the composition is made up of dissolved organics/inorganics, the droplet may require homogeneous freezing temperatures (< -37C). This possibility is not explored in this study. How to assure that this crystal (or supercooled droplet) is free of any residue/foreign substance that may trigger nucleation of ice? If the droplet could freeze at < 25C temperatures, then conclusions will change.

[Figure]

To verify this possibility an experimental evidence is needed. In response (page 3), it is mentioned that "A possible explanation for the absence of INPs are crystals formed through secondary ice formation processes.", but this is a conclusion which is drawn in this paper based on limited observations, not an explanation. Further, papers from the literature are highlighted saying that low INP concentrations compared to N_ice concentrations are observed previously, but this response does not answer the above question. There are no results regarding the nature of INPs or the freezing spectra of droplets at colder temperatures to understand this concern. My all other questions are somewhat related to this concern. Additional experimental evidence (for example as above) is needed to support the claims made in the paper.

---

## Author Response (AR1)

**Point-by-point explanation of the changes made to the manuscript in response to the comments received during the open discussion**

First of all, we would like to thank both anonymous referees for their valuable comments and suggestions. They were very helpful to us in the revision process and we think they have considerably contributed to a substantially improved, revised manuscript.

For clarity and easy visualization, the referee's comments are shown from here on in black.

> The authors' replies are in blue font with an increased indent below each of the referee's statements. Page and line numbers (in blue) refer to the original manuscript as in the online ACPD version.

> > The authors' comments about the changes made to the original manuscript are stated in green, with a further increased indent.

> > Furthermore, the relevant changed sections from the revised manuscript are copied below in red. Page and line numbers (in red) refer to the revised version of the manuscript (without track changes).

**Authors' response to Anonymous Referee #1**
**Review from Anonymous Referee #1 received and published: 28 August 2018**

General comments

The manuscript shows results of cold stage tests from samples taken at Jungfraujoch with the aim of illustrating secondary ice formation at an "individual hydrometeor level". These analyses could yield quantitative estimates of ice crystal enhancement, but the data are too few to make a publication-worthy conclusion in my opinion. The authors note that Hoffer and Braham attempted a similar per-particle analysis more than 50 years ago. Their sample size was 300 snow pellets, 150% that presented here, and they note in their abstract that "a firm statement could not be made as the number of observations is limited." The burden is on the authors to explain why it is sufficient to show ground-based data from only 10 days.

> We thank Anonymous Referee #1 for his/her assessment and valuable suggestions. In a 'short comment' we have already clarified the valid point about the sample size. We will discuss that the number of snow crystals found to have formed through heterogeneous freezing determines how firm a conclusion can be drawn and not the total number of snow crystals analysed and we will add the uncertainty of the multiplication factor during revisions. The uncertainty associated with the observed multiplication factor is about 20% (square root of 24 divided by 24). The detailed answer can be found in the short comment posted on 5th September 2018.

> > We have clarified that Hoffer and Braham (1962) could not make "a firm statement" about the multiplication factor with their sample size of 300 in the introduction by adding the following sentence:

> > However, an ice multiplication factor (i.e. the number of all ice particles divided by the primary ice particles) could not be estimated because the number of primary ice particles was zero. (page 3, lines 14-16)

In our case the number of primary ice particles was not zero but 24 out of 190. Therefore, we can derive an uncertainty of the primary ice number (√24/24) and estimate a multiplication factor (190/24). We have mentioned the uncertainty associated with our findings regarding the number of primary ice crystals in the result section with:

The uncertainty associated with the number of primary crystals in our observations is about 20% (√24/24). (page 7, lines 14-15)

Thereafter, the introduction needs to be expanded in my opinion. Right now, there is not a thorough discussion of existing literature. Approximate values and measurement techniques for INPs and IRs in mixed-phase clouds should be mentioned, in particular the abundance of measurements from Jungfraujoch (e.g. with the Ice Selective Inlet (Kupiszewski et al. 2015), the Ice Counterflow Virtual Impactor (Mertes et al. 2007), and the Horizontal Ice Nucleation Chamber (Lacher et al. 2017) as discussed in Cziczo et al. Measurements of Ice Nucleating Particles and Ice Residuals).

We will expand our introduction and refer to the mentioned studies to make clear already in the introduction the novelty of our approach. By sampling small ice particles (few tens of micron in aerodynamic diameter) at Jungfraujoch earlier studies were not able to separate ice which had formed in clouds from aerosolised parts of hoar frost growing on surrounding surfaces (Lloyd et al., 2015; Farrington et al., 2016; Beck et al., 2018). By sampling larger, regular ice particles (e.g. dendrites) we minimised the influence of local surfaces on our results (page 6 line 10-14) and can draw a conclusion regarding secondary ice formation within mixed-phase clouds.

We have expanded the introduction and mentioned the measurement techniques for INPs and ice residuals which we think are relevant for our study. This was done mainly by adding a new paragraph about ice residuals and the trade-off when selecting small pristine crystals on mountain-top stations. Also, we refer to the studies by Mertes et al. (2007), Kupiszewski et al. (2015), and Cziczo et al. (2017). Please note that we haven't mentioned Lacher et al. (2017) because they have measured INPs at much lower temperatures (~ -30°C) than those at which we have measured INPs.

While modelling studies accounting for secondary ice production can to some extent explain the observed ice crystal numbers (e.g. Sullivan et al., 2018b), field measurements have not been conclusive as to the contribution of secondary ice production mechanisms until present days. Kumai (1951, 1961) and Kumai and Francis (1962) found an insoluble particle of 0.5 to 8 µm in size in the centre of almost every one of the about 1000 snow crystals they collected. The particles they found were clay and related minerals and were assumed to have initiated the formation of the crystals. Bigg (1996) suggested to repeat the experiments by Kumai and Francis (1962) and to look at the ice nucleation properties of these particles. One reason why it can be misleading to equate ice residuals with INPs is that MPC-generated ice crystals can contain cloud condensation nuclei (CCN) which have been collected upon riming but have not acted as INPs. One possibility to overcome this issue is to sample ice residuals of freshly formed, small ice crystals (< 20 µm), which are assumed to have grown by the initial phase of vapour diffusional growth only (Mertes et al., 2007; Kupiszewski et al., 2015). On mountain-top stations, where such crystals can be collected in-cloud, however, hoar frost (cloud droplets frozen

onto surfaces) can be a strong source of small (i.e. < 100 μm) ice crystals (Lloyd et al., 2015; Farrington et al., 2016; Beck et al., 2018). Hoar frost grows in saturated conditions, breaks off when windy, and broken-off segments can become ingested into clouds and commonly mistaken for secondary ice (Rogers and Vali, 1987). Residuals in hoar frost particles are CCN that had not been activated as INPs. Only droplets freeze upon contact with an iced surface while ice particles bounce off and remain in the airflow, a principle applied in counterflow virtual impactor inlets used to separate ice from liquid in MPCs (Mertes et al., 2007). Current ice selective inlets are not able to separate primary from secondary ice (Cziczo et al., 2017). (page 2, lines 15-31).

The analyses also need to be fleshed out. A more complete picture of the meteorology could be given by including the range and variability of air temperatures and wind velocities during the sampling periods. If photos of all the crystals were taken with a high-quality camera, some of these should be shown. Is there a more rigorous means of classifying the crystals than what is "considered to be planar, branched"? If the size of the crystals was measured with ImageJ, could some of these statistics also be presented? In Section 2.3, there is also a mention of rime analyses with a second cold plate, but it was not clear to me how this fit in. The results shown in Figures 2 and 3 are from pristine, unrimed dendrites, right?

We will provide the range and variability of air temperatures and wind velocities in a revised version. We are ready to show representative pictures of the crystals that were taken in the main paper or all pictures as supporting information. As mentioned on page 3 line 31, we classified the crystals by habit and riming degree using the 'global classification scheme'. There are some variations in crystal shape. We considered as planar, branched crystals, crystals that can be classified into the following classes: R1c, R2c, R3a, R4a, P4, P3, P2 according to Kikuchi et al. (2013) (see first paragraph in Section 3; please note that the paper by Kikuchi et al. contains representative images of all the crystal classes mentioned in our paper). Figures 2 and 3 show the results of the 190 crystals. A large fraction of them are 'rimed dendrites' or 'densely rimed dendrites' (see page 5 line 6). Rime itself was analysed to determine the fraction of analysed crystals that possibly had scavenged through riming an INP active at -17 °C or warmer. This fraction was smaller than 1% (page 6, lines 32-33).

We have extended the methods section in various ways. We have provided the standard deviations of the daily temperatures and wind velocities in Table 1 and the standard deviation of the mean air temperature and wind velocity during the sampling periods.

The mean air temperature at the station during the sampling periods was -11.0 °C (±3.6) and the mean wind velocity was 9.1 m s$^{-1}$ (±3.9). (page 4, lines 9-10)

We have added example images of the analysed dendrites in the supplement (see Fig. S3 in the revised version).

[Figure]

**Figure S3.** Examples of images of the analysed dendrites taken by macro (1:1) photography. Crystals of which the residues immersed in a water droplet froze above -20 °C have this freezing temperature added in the upper left corner of their image. Each image shows an area of 8.6 mm x 6.4 on the black surface where the crystals had been collected.

We have clarified our selection criteria of the analysed crystals in section 2.2.

**2.2 Single crystal selection and analysis**

[revised manuscript text omitted]

With additional data and stronger analysis, more could be gleaned from this study. If the cold plate measurements are subject to any contamination, then 12.6% of the droplets refreezing is actually an overestimation. And a limited crystal geometry has been used to define secondary ice; at -15°C and lower supersaturations, other geometries are possible. So perhaps the multiplication factor of 8 is more of a lower bound. Quantitative estimates of this factor are needed for models, and field measurements at the hydrometeor level, rather than the bulk cloud level, are a new, if labor intensive, technique.

With the same number of control droplets as droplets from crystals we assessed potential contamination. For temperatures at which the analysed crystals had formed (-12 °C to -17 °C) only 0.5% (1 in 190) of the droplets were contaminated. Indeed, at lower

supersaturations other crystal geometries are possible at around -15 °C. However, as we were sampling within mixed-phase clouds, we were always within highly supersaturated conditions. We would like to recall that our aim was to find reliable evidence for secondary ice formation at around -15 °C in clouds. For this reason, we had to exclude as rigorously as possible the influence of secondary ice formed and aerosolised from local surfaces (e.g. hoar frost). This requirement called for selecting crystals with a regular shape that forms in clouds and a size large enough to tell they have not grown from splinters emitted locally (see page 6, lines 5-14). We agree that the estimated ice multiplication factor may therefore be a lower bound, a point we will make clear when revising the manuscript.

Our multiplication factor is based on the total number of crystals analysed and the number of ice crystals that froze at -17 °C and warmer (INP$_{-17}$). At this temperature and above only one of 190 control droplets froze. Furthermore, we have clarified that the number of crystals that froze at -25 °C is not relevant for the multiplication factor at around -15 °C. However, it was useful to determine the fraction of rime associated with single crystals, for which the number of frozen control droplets (at -25 °C) were taken into account (see Eq. 1).

The presence of an INP active at -17 °C and warmer (INP$_{-17}$) was taken as evidence for the tested dendrite to have been generated through primary ice formation. Nevertheless, extending the drop freeze assay down to -25 °C is useful to determine the fraction of rime associated with single crystals (see Sect. 2.3). (page 5 lines 23-25).

The ice multiplication factor we found is only valid for dendrites. Therefore, we have added the following sentence in the conclusion:

However, we do not know whether the ice multiplication factor we found for dendrites is the same for other crystal habits found in the same MPCs. (page 9, lines 7-8)

Specific comments

Page 1, Lines 18-20: The conclusion that "secondary ice can be observed at temperatures around -15°C" is not an especially compelling one, given that many previous studies have already shown this. Is there a hypothesized mechanism? Or was observed multiplication factor higher under certain conditions?

As far as we know, no previous study has directly observed secondary-produced ice at around -15 °C in natural mixed-phase clouds. What has been reported were large discrepancies between number concentrations of ice crystals and INPs. We could only speculate which mechanisms is responsible for the secondarily produced ice by referring to the papers by Field et al. (2017) and Sullivan et al. (2018), both cited in the manuscript. As shown in Figure 3 the daily fraction of primary crystals was relatively constant and varied around the mean value of the pooled data. When considering the uncertainty of those days where we had at least four primary crystals (black dots), their means are not distinguishable from the pooled data (mean +/- standard deviation of the pooled data).

We have kept the conclusion as it was as we think that this is a new type of evidence for secondary-produce dendrites at around -15 °C. However, the referee's comment has led us to change the title of the manuscript to:

New type of evidence for secondary ice formation at around -15 °C in mixed-phase clouds

Furthermore, we have put more emphasis on the novelty of our study in the abstract.

The novelty of our approach lies in comparing the growth temperature encoded in the habit (shape) of an individual crystal with the activation temperature of the most efficient INP contained within the same crystal to tell whether it may be the result of primary ice formation. (page 1, lines 14-16)

We have added a sentence in the conclusion section to clarify that based on our results we do not know which mechanisms were responsible for the secondary ice production.

No conclusion regarding the process of secondary ice formation can be drawn from our observation. (page 9, lines 8-9)

Page 1, Line 23 – "These freezing pathways" as there can be contact or deposition or immersion freezing.

Correct, thank you.

The sentence has been changed to:

In mixed-phase clouds (MPCs), heterogeneous freezing is expected to generate ice crystals, but also secondary ice production mechanisms can enhance the ice crystal number concentration (Cantrell and Heymsfield, 2005). (page 1, lines 25-27)

Page 1, Line 26 – A few additional, more recent observations might be cited. For example, Lasher-Trapp et al. JAS [2016], Ladino et al. GRL [2017], and Jackson et al. ACPD [2018].

We will add them.

We have added them see page 2, line 3 and page 3, line 1 in the revised manuscript.

Page 2, Line 19 – For completeness, you could mention the correction of such shattering artifacts in more recent data by inter-arrival time algorithms and K-tip probes.

Thank you for mentioning; we will do that.

Not applicable anymore. The related sentence fell victim to the substantial revision of the manuscript.

Page 2, Line 22 – I would define rime when you first discuss rime splintering above in Lines 4-5.

O.k., we will define rime there.

We have done so.

For example, secondary ice crystals can result from rime splinters that are released upon riming (i.e. supercooled cloud droplets that freeze upon contact with a solid

hydrometeor) of ice crystals at temperatures between -3 °C and -8 °C (Hallett and Mossop, 1974; Jackson et al., 2018). (page 2, lines 1-3)

Section 2.2 and Figure 2 – The authors have taken a number of concerns about cold-stage measurements into consideration with their setup, which I appreciate. I would cite Tobo 2016 for the use of a semi-sold, hydrophobic substrate, and you might mention the possibility that INP settle out or aggregate within your large-volume droplet [e.g., Emerstic et al. 2015 ACP]. I am still concerned, however, that 20% of the control droplets have frozen by -25°C, almost 10°C above the threshold temperature for homogeneous freezing. Could the estimated enhancement factor be adjusted to account for these "false positives"?

The estimated enhancement factor relates to the temperature window in which the collected ice crystals were likely to have formed (-12 °C to -17 °C). In this temperature window we had only one false positive in 190 tested controls. The number of droplets frozen by -25 °C only plays a role when estimating the average mass of rime associated with a single crystal. In this estimate we have accounted for the false positives (subtracted frozen controls from frozen droplets; please see page 6, line 27).

We have cited Tobo (2016) and Polen et al. (2018) regarding the cover of petroleum jelly.

With a fine brush, two crystals are transferred onto the cold-stage surface thinly covered with Vaseline® petroleum jelly (Tobo, 2016; Polen et al., 2018) before being analysed within the next minutes (Fig. 1a). (page 5 lines 9-10)

That 20% of the control droplets are frozen at -25 °C is not unusual for drop freezing assay. It is rather low compared to the results obtained by Polen et al. 2018.

A frozen fraction of 21% of the control droplets at -25 °C is a rather low fraction compared to the results with pure water droplets (1 µL) on a Vaseline-coated substrate presented recently by Polen et al. (2018). (page 7, lines 7-9)

Page 4, Line 1 – I would add a sentence that summarizes what this 'global classification scheme' is because it is not so widely used, as far as I know.

We will add a sentence that summarizes the global classification scheme.

We have added a sentence.

Images were later analysed more exactly for the habit, including the degree of riming both categorized according to the latest ice crystal classification scheme, as presented by Kikuchi et al. (2013). The scheme catalogues solid precipitation particles into a total of 121 categories and provides for each category a representative image. (page 4 lines 25-29)

Page 4, Line 27 – Is there a reason that the "custom-built cold stage" used for single crystal analysis was not also used for the rime?

Unlike traditional cold plate systems, the custom-built cold stage is mainly made to be easily field transportable to remote locations. It was however not used for the rime samples as it has a rather small surface (surface diameter of 18 mm, page 4 line 5). Analysing rime with it would have led to less measurement time for the single crystals. Our goal was to get as much measurement time for the single crystals as possible. This requires a cold stage which

is ready to be used when the specific type of crystals precipitate. The second cold stage i.e. the NOAA Drop Freezing Cold Plate has a larger surface, therefore more droplets can be placed on it, which is convenient for the rime analysis. Please note that the NOAA Drop Freezing Cold Plate requires an external refrigerated circulation bath, lined power and is relatively large. We could not put it into the anteroom and analyse single crystals. We used the most suited cold stage type available for each sample type. We will add the reason why we used two different cold plate systems in the revised manuscript and discuss whether the results of both plates are comparable.

We have listed the reasons why the custom-build cold stage was not also used for the rime in the section 2.3. Please note that the naming "NOAA Drop Freezing Cold Plate" has been changed to simply "drop freezing cold plate".

The main reason for the use of a second cold-stage was to ensure that the custom-build one was always ready for single crystal analysis in case dendrites were precipitating. Other than that, the drop freezing cold plate has a larger surface on which more droplets can be analysed at a time making it more suitable for rime analysis. However, it also requires an external refrigerated circulation bath, lined power and it is relatively large, making it impossible to put it into the anteroom and to analyse single crystals. (page 6, lines 15-19)

Page 4, Lines 28-29 – I am not sure what is meant by "droplets of molten rime". You are melting the aggregation of frozen droplets and then refreezing them upon a cold plate? Or somehow separating the droplets within a single aggregate? Please clarify here.

Indeed, we are melting the aggregation of the frozen rime droplets and then refreezing them. We will clarify this in a revised manuscript.

We have rephrased this part.

However, rime samples were melted and portioned with a sterile syringe into 2.5 µL droplets and analysed with a drop freezing cold plate following the description in Creamean et al. (2018a). (page 6, lines 10-12)

Page 5, Lines 19-21 – Measurement uncertainty and / or variability for this estimate needs to be included.

Measurement uncertainty will be included on page 5, lines 19-21.

We have included the uncertainty.

The uncertainty associated with the number of primary crystals in our observations is about 20% ($\sqrt{24}/24$). (page 7, lines 14-15)

Page 5, Line 32-Page 6, Line 1 – The mention of INP from soils does not seem particularly relevant to me, as those will not be the INP source at Jungfraujoch.

This sentence presents one of three examples from the literature that illustrate the ice nucleation temperature stability during repeated melting and freezing and therefore we think that it is worthwhile to mention it. Besides that, we think that aerosolised soil particles, or soil dust, potentially emitted from fields in northern Italy, southern France, southern Germany, and the Swiss Plateau might make a relevant contribution to INPs active at relatively high temperatures (i.e. > -17 °C) at Jungfraujoch. Note that the most prominent

particle classes (reflecting particles in the size range between 0.5 um and 5 um) determined at Jungfraujoch were carbonaceous particles (Hinz et al., 2005). Furthermore, most of the fields within the fetch of Jungfraujoch are not covered by snow during winter and wind blown dust emissions are relatively high during that season in Europe (Korcz et al., 2009).

We think that it is still worth mentioning it because of the above mentioned reasons and therefore we kept it.

Page 6, Lines 5-14 – Blowing snow is a very important consideration here, given several existing studies on this mechanism at Jungfraujoch. You are considering pristine dendrites here, right? Otherwise, there is the potential for riming growth, not just depositional growth.

We are considering planar, branched crystals including rimed crystals. This is the reason why we have also analysed the INP spectra of rime itself. Our results show that less than 1% of the analysed crystals may have scavenged an INP active at a similar temperature as the INP which might have catalysed the formation of a dendrite (page 6 lines 23-34).

We have emphasized which crystals we have selected in section 2.2.

Among a usually wide variety of shapes and sizes precipitating onto the plate we selected what we considered to be single, planar, branched or dendritic ice crystals (from here on "dendrites"), which can safely be assumed to have grown within MPCs at temperatures around -15 °C (Nakaya, 1954; Magono, 1962; Magono and Lee, 1966; Takahashi et al., 1991, Takahashi, 2014; Libbrecht, 2017). Our selection criteria exclude hoar frost particles which might have been generated by local surface sources around the station (Llyod et al., 2015; Farrington et al., 2016; Beck et al., 2018). Rime on selected crystals is of little concern in our approach and was accounted for (see Sect. 2.3). (page 4, lines 15-20)

As they were not necessarily pristine, we have accounted for riming as described in section 2.3.

A rimed ice crystal has collected liquid cloud droplets, each of them containing a CCN that may cause freezing of the droplet containing the residuals of this crystal. Such a CCN may be activated on the cold-stage as INP (from here on: scavenged INP), although it had not initiated the formation of the collected dendrite. The median concentration of INPs active at -25 °C or warmer (INP-25) was determined for bulk rime samples collected on impactor plates (concrime) and used to estimate the mass of rime associated with a single dendrite (m): (see Eq.1). This step was necessary to estimate the contribution of scavenged $INP_{-17}$ representing false positives of primary ice crystals in our results. (pages 5-6, lines 28-4)

Table 1 – For periods that last as much as 14 hours, it would be more rigorous to give mean and standard deviation for values like air temperature / wind velocity since a single value will not be characteristic. Are there are any vertical wind measurements?

O.k., we will add standard deviations for air temperature and wind velocity. No vertical wind measurements were taken though.

We have added the standard deviations for air temperature and wind velocity (see Table 1.)

Table 1. Sampling periods including the date and the time span, numbers of analysed crystals (n), mean air temperature (T) (and standard deviation), mean wind velocity (u) (and standard deviation) and mean wind direction (dd) at Jungfraujoch; mean height of the station above cloud base (zB) and estimated mean cloud base temperature (CBT).

| Date | Time span | n | T | u | dd | zB | CBT |
|---|---|---|---|---|---|---|---|
| dd/mm/yyyy | UTC | - | °C | m/s | - | m | °C |
| 15/02/2018 | 07:30 - 21:50 | 38 | -7.0 (0.8) | 13.5 (2.1) | NW | 944 | 0.1 |
| 16/02/2018 | 09:30 - 16:30 | 29 | -8.7 (0.2) | 9.0 (2.4) | NW | 1239 | 0.6 |
| 17/02/2018 | 09:40 - 23:40 | 42 | -8.5 (1.7) | 5.8 (1.9) | NW | 693 | -3.3 |
| 23/02/2018 | 10:30 - 21:20 | 20 | -14.8 (0.6) | 11.9 (1.6) | SE | 365 | -12.1 |
| 06/03/2018 | 12:20 - 19:20 | 14 | -13.1 (0.1) | 5.5 (0.8) | NW | 1284 | -3.4 |
| 07/03/2018 | 08:00 - 16:40 | 23 | -15.7 (0.8) | 4.5 (2.6) | NW | 1001 | -8.2 |
| 10/03/2018 | 09:30 - 12:50 | 11 | -6.8 (0.3) | 5.1 (1.3) | E | 196 | -5.4 |
| 11/03/2018 | 15:40 - 17:00 | 6 | -9.8 (0.1) | 13.1 (1.4) | SE | 1485 | 1.3 |
| 12/03/2018 | 09:10 - 11:10 | 12 | -11.4 (0.1) | 6.2 (0.7) | NW | 878 | -4.8 |
| 22/03/2018 | 15:50 - 22:30 | 34 | -15.2 (1.2) | 12.4 (1.5) | NW | 1079 | -7.1 |

**Authors' response to interactive comment from Anonymous Referee #1**
**Interactive comment received and published: 10 September**

Thanks very much for your response concerning the uncertainty in the ice multiplication factor estimate given your sample. My concern is more related to the uncertainty in the representativeness of your sample for the population. Assuming that "the population" here is ice in a mixed-phase cloud, then you can estimate the population size ..say conservatively that the cloud is 2 km deep and has an equivalent radius of 3 km, then it already has a volume on the order of 10^10 cubic meters. Even if the ice crystal concentration in the cloud is only a crystal per cubic meter, you have sampled a very small portion of the population for which you are making a conclusion. This is how I am thinking, but I understand that there are all sorts of subtleties related to representativeness and that your collection process is laborious, so let us see what other reviewers say.

The number of crystals we have sampled and analysed is indeed a very small fraction of all crystals in all clouds that have passed Jungfraujoch between 15 February and 22 March 2018. If we had sampled the crystals from a small fraction of a cloud volume and would extrapolate our findings to a much larger volume in which primary and secondary crystals are very heterogeneously distributed, we would face a problem. However, we have sampled crystals on 10 days in different clouds spread over a period of 36 calendar days. To estimate the total path of clouds crossing Jungfraujoch during our sampling events, we can multiply the sampling duration of an event with the mean wind speed (values in Table 1). By doing so and taking the sum of all the sampling events, we get a total path of 2368 km within clouds along which we have taken our samples. We think that this is a representative distance of cloud passage and thus a representative sample for this year's winter clouds at Jungfraujoch. Figure 2 shows that the ice multiplication factor for individual days is similar to the mean of the pooled data, considering the larger uncertainty of daily estimates as compared to the estimate for the whole period. Hence, the average of the period is not subject to bias of a single day with substantially different multiplication factor from the rest of the days.

We have mentioned that we have taken samples from a long pathlength within different clouds.

They had been collected from a pathlength of 2368 km through a large number of MPCs from different wind directions (sum of sampling duration multiplied by average wind speed; see Table 1). (page 6, lines 25-26)

**Authors' response to Anonymous Referee #2**
**Review received and published: 11 September 2018**

Authors present the experimental work where they collected the snow crystals, melted the crystals and visually observed the freezing of the crystal droplet. These results were used to understand more about the secondary ice formation and ice multiplication factors. These questions are challenging, and the community needs an understanding of these cloud processes for better representation in the cloud model. However, this study lacks appropriate experimental technique/methodology to answer these questions, and for this reason, the paper is not ready for the publication. I'm not sure if the major review could improve the paper further as substantial experimental work is involved. There are a number of issues in the present experimental study.

We thank Anonymous Referee #2 for openly sharing his/her ideas on our recent manuscript. We agree that the questions about secondary ice formation and multiplication factors are challenging and that the community needs to answer them to improve cloud parametrizations in models. There are different approaches to answer these questions. The work presented here addresses them by applying an unconventional, new method. The study combines the growth temperature encoded in the habit of snow crystals with a drop freezing assay and thus complements previous observations of secondary ice formation. Our experimental technique is appropriate for detecting insoluble ice nucleation particles (INPs) in single crystals and enables us to estimate with an uncertainty of about 20% the lower bound of the ice multiplication factor in clouds during our sampling campaign at Jungfraujoch. Herewith, we would like to dispel the referee's doubts and elucidate how we will make use of the referee's comments in a revised version of the manuscript. Furthermore, we are confident that our manuscript constitutes a valuable contribution to ACP and we appreciate the opportunity to openly stand up for and constructively discuss our work.

We have clarified in the revised version of the manuscript how we addressed the quantification of secondary ice in MPCs by applying a new and appropriate experimental methodology. The changes made with regard to the general comment above are too numerous to list here at once. They are indicated in the more detailed sections below.

**Section1**

If no INP was observed within the crystal, it does not mean that crystal was formed through secondary ice formation mechanism. It is possible that a INP may have induced nucleation of ice, and still while INP is floating within the atmosphere may have detached from the ice crystal because the crystal evaporated or through some turbulent process. Now, this crystal when sampled had no INP.

We are not aware of any literature describing the mechanisms to which this statement could refer to. Does Anonymous Referee #2 have supportive evidence for ice crystals losing their INP through evaporation, sublimation, or through "some turbulent process" in the atmosphere that are resulting in ice particles without INP?

In the atmosphere, ice nucleation has been observed at temperatures warmer than that of homogeneous freezing (Ansmann et al., 2005). Four main pathways of heterogeneous freezing

have been identified: contact, deposition, condensation, and immersion freezing (Pruppacher and Klett, 1997). In our study, we investigated freezing through the immersion freezing mechanism. Immersion freezing refers to the initiation of ice nucleation by a solid and insoluble INP immersed in a water droplet. To our understanding, the immersed INP will catalyse an initial crystal, in which the INP is embedded. This initial crystal then grows through vapour deposition. In this process, the INP in the initial crystal will increasingly become encased in ice that grows thicker around it. If this crystal then begins to sublimate, the ice covering the initial droplet surrounding the INP will become thinner again, which we expect to evolve rather uniformly from the outside (i.e. edge of the crystal) towards the inside (i.e. initial droplet that froze by immersion). The INP will be released from the ice only once the ice of the very initial frozen droplet has sublimated, resulting in an INP without ice, but not in ice without an INP.

Besides that, we are also not aware of observations that show how "some turbulent processes" may detach the INP from a crystal. How should the INP get out of the crystal structure? Is there at all relevant turbulent friction at the submillimetre-scale in the free atmosphere? We would however be happy to discuss such mechanisms in our manuscript if they have a theoretical or observational basis.

As already mentioned in our published reply above, we are not aware of any literature describing the mechanisms to which this statement could refer to and therefore we were not able to change the revised manuscript accordingly.

**Section 2**
It is also possible that INP is present, but was deactivated while it went transformation (change in physical and chemical properties) during sampling, heating or droplet preparation. There are numerous studies in the literature that discusses the deactivation of INP. Such discussion is missing.

Indeed, studies exist that discuss the deactivation of INP during transformation. In our experiment, the crystals were sampled below melting temperatures, and melted or "heated" to between +1 °C and +5 °C (page 4, line 12) before being analysed within the next minutes. It is not unusual to store INPs in water at +4 °C for several hours before analysis (e.g. Wilson et al., 2015). Studies reporting deactivation through heating typically refer to heating temperatures close to the boiling point of water (e.g. Christner et al., 2008).

There is also convincing evidence in the literature that INPs, which are active at temperatures relevant for our study, can be repeatedly activated, going through multiple cycles of freezing and melting. We have discussed and referred to these studies on page 5 line 30 to page 6 line 3. Furthermore, we have clearly formulated that our findings are based on the assumption that the cited evidence also applies to our samples, see page 5 line 23.

Several laboratory studies have investigated the role of coating of mineral dust particles and the related changes in ice nucleation efficiency (e.g. Knopf and Koop 2006, Cziczo et al., 2009; Kanji et al., 2018). Soluble coating or soluble INPs could be altered through melting or droplet preparation. However, the work presented here is not investigating the effect of soluble coating and neither of soluble INPs. Soluble INPs probably do not play a role at temperatures warmer than about -27 °C (Knopf et al., 2018, see their Fig. 5). Based on the referee's comment, we will emphasize in a revised version, that we are focusing on insoluble INPs in dendrites that can be activated through immersion freezing at temperatures above -17 °C for at least two freezing cycles (one when forming the crystal and one when doing the measurement).

We have clarified that we have only tested the crystals on insoluble INPs through immersion freezing.

After selecting the crystals, we tested them for the most efficient insoluble INP they contain that can be activated through immersion freezing using a custom-built cold-stage (Fig. 1; more details in supplement). (page 4, lines 30-31)

Furthermore, we have added a note that the temperature at which we have melted the samples is a common temperature at which samples can be stored for a few hours before for INP analysis

At the transfer of the crystals, the surface of the stage was at a temperature between +1 °C and +5 °C, which is a common temperature range to store INPs in water for several hours before analysis (e.g. Wilson et al., 2015). (page 5, lines 11-13)

Also, we have added in the conclusion that our findings are based on a number of assumptions in the conclusion section, which were discussed before.

The habit of a planar, branched ice crystal, growing exclusively around -15 °C, enables the verification of whether it derived from primary or secondary ice formation based on a number of reasonable assumptions. (page 9, lines 2-3)

**Section 3**
Experiments are needed that investigate the ice nucleation efficiency of crystal melted droplets up to -37 degC (below this temperature homogeneous freezing is the dominant mode of ice nucleation) to understand more about the insoluble INPs, but for soluble INPs experiments should be investigated at homogeneous freezing temperatures too. Without such results, the conclusions regarding secondary ice formation cannot be inferred.

Heterogeneous freezing at temperatures below -25 °C and homogenous freezing at even colder temperatures are certainly important topics of research, especially when investigating cold mixed-phase clouds or cirrus clouds. Observations have shown that an overwhelming majority of ice particles originate from supercooled liquid clouds at temperatures > -27 °C, which strongly suggests that the initial process of ice formation in mixed-phase clouds > -27 °C occurs through immersion freezing (Westbrook and Illingworth, 2011). Therefore, we assume that homogeneous freezing does not play an important role in mixed-phase clouds surrounding Jungfraujoch during our campaign where temperatures were clearly higher (see Table 1). Further, every experimental study has a limited parameter space. We set the frame for our study in the second part of the introduction. Briefly, our objective was to detect the presence of INPs active at around -15 °C in dendrites, which typically grow around that temperature. By investigating ice nucleation down to -25 °C we already expanded our measurements well beyond the necessary to answer the question to what proportion dendrites are the result of primary ice formation.

We have clarified why immersion freezing experiments down to -25°C are a suitable way to address the question addressed in the presented study (ice multiplication at around -15 °C). In addition to the two short paragraphs below, please also see the substantially revised Sect. 2.3 in the new manuscript.

Observations have shown that an overwhelming majority of ice particles originate from supercooled liquid clouds at temperatures > -27 °C, which strongly suggests that the initial process of ice formation in MPCs > -27 °C occurs through immersion freezing (Westbrook and Illingworth, 2011). (page 5, lines 2-4)

The presence of an INP active at -17 °C and warmer (INP$_{-17}$) was taken as evidence for the tested dendrite to have been generated through primary ice formation. Nevertheless, extending the drop freeze assay down to -25 °C is useful to determine the fraction of rime associated with single crystals (see Sect. 2.3). (page 5, lines 23-25)

**Section 4**

Supporting experiments are needed to say why there was no INPs present (page 5 line 14). It would be just that the limitation of the experimental setup. In this study, the sample collection onto the cold stage is not done in clean air conditions. It is possible that crystals were contaminated with room air particles. Further, it is possible that these particles may have induced nucleation of ice but not the primary INP (the first INP that was responsible for freezing the droplet in the atmosphere before sampling). Without knowing the composition of residue it is difficult to infer which INP (primary or room air particulates) was responsible for freezing.

Indeed, it is very important in a first step to avoid contamination as much as possible and in a second step to quantify it. We examined contamination with control droplets of molecular grade water (blanks). If contamination, including deposition of INP from the room air would have been a problem, we would have seen it in the freezing of control droplets. As shown in Fig. 2 and discussed in the text, of 190 control droplets only one froze within the temperature range where the analysed crystals may have formed (-12 °C to -17 °C). Deposition of "room air-INPs" is only one out of several possible reasons why this control droplet may have frozen. Another reason could have been surface contamination of the cold stage. Please note that the control droplets were exposed to the same room air during the same time as were our sample. Thus, even without knowing the composition of residue, we can show, with the results of the control droplets, that INPs deposited from room can not have been responsible for the freezing of the crystal droplets.

It is not a limitation of our experimental setup that no INP active around -15 °C was found in a large proportion of the analysed dendrites. A possible explanation for the absence of INPs are crystals formed through secondary ice formation processes. Our results are consistent with findings and conclusions from other studies (page 1 line 26). Several studies measured much lower INP concentrations than ice crystal number concentrations in clouds by using different approaches and measurement techniques from ours.

We have added a section in the introduction clarifying why the composition of ice residuals will not help us to answer our question.

One reason why it can be misleading to equate ice residuals with INPs is that MPC-generated ice crystals can contain cloud condensation nuclei (CCN) which have been collected upon riming but have not acted as INPs. One possibility to overcome this issue is to sample ice residuals of freshly formed, small ice crystals (< 20 µm), which are assumed to have grown by the initial phase of vapour diffusional growth only (Mertes et al., 2007; Kupiszewski et al., 2015). On mountain-top stations, where such crystals can be collected in-cloud, however, hoar frost (cloud droplets frozen onto surfaces) can be a strong source of small (i.e. < 100 µm) ice crystals (Lloyd et

al., 2015; Farrington et al., 2016; Beck et al., 2018). Hoar frost grows in saturated conditions, breaks off when windy, and broken-off segments can become ingested into clouds and commonly mistaken for secondary ice (Rogers and Vali, 1987). Residuals in hoar frost particles are CCN that had not been activated as INPs. Only droplets freeze upon contact with an iced surface while ice particles bounce off and remain in the airflow, a principle applied in counterflow virtual impactor inlets used to separate ice from liquid in MPCs (Mertes et al., 2007). Current ice selective inlets are not able to separate primary from secondary ice (Cziczo et al., 2017). (page 2, lines 20-31)

We have accounted for false positive through riming as described in section 2.3.

A rimed ice crystal has collected liquid cloud droplets, each of them containing a CCN that may cause freezing of the droplet containing the residuals of this crystal. Such a CCN may be activated on the cold-stage as INP (from here on: scavenged INP), although it had not initiated the formation of the collected dendrite. The median concentration of INPs active at -25 °C or warmer (INP$_{-25}$) was determined for bulk rime samples collected on impactor plates (conc$_{rime}$) and used to estimate the mass of rime associated with a single dendrite (m): (see Eq.1). This step was necessary to estimate the contribution of scavenged INP$_{-17}$ representing false positives of primary ice crystals in our results. (pages 5-6, lines 28-4)

Also, we have changed the following sentence to more precisely state how we avoided contamination.

Shortly after the cold-stage temperature reached a value below the surrounding air temperature, we covered it with a transparent hood to minimise the chance for contamination from the environment surrounding the droplets and to prevent condensation on the cold-stage (Polen et al., 2018). (page 5, lines 18-21)

Furthermore, we have added supporting literature which found larger ice crystals number concentrations than INP number concentration:

Most such studies report large discrepancies between measured INPs and ice crystal numbers (e.g. Hobbs and Rangno, 1985; Lasher-Trapp et al., 2016; Ladino et al., 2017; Beck et al., 2018) the latter being several orders of magnitudes higher than the former. (page 2-3, lines 34-2)

**Section 5**
It is not clear how section 2.3 supports the secondary ice formation analysis. Details such as validation and performance calibration of the cold stage (shown in Fig 1) under different temperature and humidity conditions are missing.

The majority of analysed crystals were rimed. Rime could have added INPs active at around -15 °C to initial crystals (page 2 line 23-24). Therefore, we analysed not only (rimed) crystals but also rime itself (method in section 2.3). Our results show that riming had only a very minor influence on our results (page 6 line 23-34).

The cold stage was used to test for INPs in immersion freezing mode. Details of the cold stage as well as calibration can be found in the supplement, including the result of tests at a range of temperatures. We are not sure why we should perform validation and calibration at

different humidity conditions. These would play a role only, if we would study deposition or condensation freezing.

We have extended section 2.3 in order to clarify why the collection and analysis of rime samples was necessary and how it supports our study.

**2.3 Accounting for riming**

A rimed ice crystal has collected liquid cloud droplets, each of them containing a CCN that may cause freezing of the droplet containing the residuals of this crystal. Such a CCN may be activated on the cold-stage as INP (from here on: scavenged INP), although it had not initiated the formation of the collected dendrite. The median concentration of INPs active at -25 °C or warmer ($INP_{-25}$) was determined for bulk rime samples collected on impactor plates ($conc_{rime}$) and used to estimate the mass of rime associated with a single dendrite ($m$):

$$m \,[g \; rime \; crystal^{-1}] = \frac{\ln((1-FF_{crystal})^{-1}))}{conc_{rime}} \; [\, INP_{-25} \; crystal^{-1}/INP_{-25} \; g^{-1} \; bulk \; rime \,], \qquad (1)$$

with $FF_{crystal}$: the frozen fraction of $INP_{-25}$ in the analysed dendrites (after subtracting the control).

This step was necessary to estimate the contribution of scavenged $INP_{-17}$ representing false positives of primary ice crystals in our results. They were estimated from the average mass of rime associated with a single dendrite (Eq. 1) and the concentration of $INP_{-17}$ within the independent rime samples as described next.

Independent rime samples were collected with a plexiglass impactor plate (Lacher et al., 2017) suspended on the railing of the terrace at Jungfraujoch for a few to several hours (~1-13h). In total, 30 samples of aggregated rime droplets were collected between 15 February and 11 March. The freezing experiments of the rime samples were done with a drop freezing assay similar to the set up described above which was used for the single crystal analysis. However, rime samples were melted and portioned with a sterile syringe into 2.5 µL droplets and analysed with a drop freezing cold plate following the description in Creamean et al. (2018a). Of each sample 300 droplets were cooled until all droplets were frozen. The cumulative number of INPs active at a certain temperature (with a temperature interval of 0.5 °C) was calculated by taking into account the observed numbers of frozen droplets at a temperature, the total number of droplets and the analysed volume of sample (Vali, 1971b). The main reason for the use of a second cold-stage was to ensure that the custom-build one was always ready for single crystal analysis in case dendrites were precipitating. Other than that, the drop freezing cold plate has a larger surface on which more droplets can be analysed at a time making it more suitable for rime analysis. However, it also requires an external refrigerated circulation bath, lined power and it is relatively large, making it impossible to put it into the anteroom and to analyse single crystals. (pages 5-6, lines 27-19)

**Section 6**

Any results from previous studies who had attempted to study secondary ice formation should be shown in Figure 2 and 3.

It would make sense to compare our results with previous studies. However, the results of previous studies are based on completely different approaches. Their results are not directly comparable to ours. One of the main differences is that we have analysed relatively large snow crystals (several millimetres in diameter) to make sure our results are not influenced by local surface sources of secondary ice formation. We will discuss differences regarding results and methodology between previous studies and this study in more detail in a revised version of our manuscript.

We clarified the difference between our and previous studies mainly in the introduction and in the methods section.

**Section 7**

Discussion regarding nature of INP is missing. What are their composition and size? One should use Ice-CVI (Mertes et al 2007) to sample only ice crystals, sublimate/evaporate these crystals, count the residues and investigate the ice nucleation propensity of a single residue. By comparing inlet ice crystal and residue concentrations one can infer some understanding regarding secondary ice formation.

Mertes et al. (2007) sampled very small ice particles, between 5 and 20 micron (aerodynamic diameter). Lloyd et al. (2015) concluded for Jungfraujoch that "hoar frost crystals generated at the cloud enveloped snow surface could be the most important source of cloud ice concentrations." The same may apply to other mountain stations (Beck et al., 2018). Therefore, repeating the experiments of Mertes et al. (2007) would tell us mainly about ice residues in hoar frost particles generated by local surfaces. This is not what we are interested in. We would like to know more about secondary ice formation in mixed-phase clouds themselves. This is the reason why we have sampled larger crystals with a regular shape that are unlikely to have resulted from surface processes and tested these crystals for the presence of INPs active within the temperature range they typically form.

We could not use an Ice-CVI to determine the multiplication factor, because such an inlet is not able to separate primary from secondary ice. We have clarified this in a new paragraph in the introduction.

One reason why it can be misleading to equate ice residuals with INPs is that MPC-generated ice crystals can contain cloud condensation nuclei (CCN) which have been collected upon riming but have not acted as INPs. One possibility to overcome this issue is to sample ice residuals of freshly formed, small ice crystals (< 20 μm), which are assumed to have grown by the initial phase of vapour diffusional growth only (Mertes et al., 2007; Kupiszewski et al., 2015). On mountain-top stations, where such crystals can be collected in-cloud, however, hoar frost (cloud droplets frozen onto surfaces) can be a strong source of small (i.e. < 100 μm) ice crystals (Lloyd et al., 2015; Farrington et al., 2016; Beck et al., 2018). Hoar frost grows in saturated conditions, breaks off when windy, and broken-off segments can become ingested into clouds and commonly mistaken for secondary ice (Rogers and Vali, 1987). Residuals in hoar frost particles are CCN that had not been activated as INPs. Only droplets freeze upon contact with an iced surface while ice particles bounce off and remain in the airflow, a principle applied in counterflow virtual impactor inlets used to separate ice from liquid in MPCs (Mertes et al., 2007). Current ice selective inlets are not able to separate primary from secondary ice (Cziczo et al., 2017). (page 2, lines 20-31).

**Authors' response to Anonymous Referee #2, RC4**
**Review received and published: 3 October 2018**

Thanks for providing more information about these experiments. However, authors do not address the concerns that are outlined. I will describe one example here. One of the conclusions of this study (page 5 main paper) is that if no INP was found in a crystal – this crystal was categorized as formed through the process of secondary ice formation. This is based on an observation that this particular crystal (now supercooled droplet) did not freeze until -25C. However, it is possible that this droplet may freeze at colder temperatures than -25C, and if the composition is made up of dissolved organics/inorganics, the droplet may require homogeneous freezing temperatures (< -37C). This possibility is not explored in this study. How to assure that this crystal (or super- cooled droplet) is free of any residue/foreign substance that may trigger nucleation of ice? If the droplet could freeze at < 25C temperatures, then conclusions will change. To verify this possibility an experimental evidence is needed. In response (page 3), it is mentioned that "A possible explanation for the absence of INPs are crystals formed through secondary ice formation processes.", but this is a conclusion which is drawn in this paper based on limited observations, not an explanation. Further, papers from the literature are highlighted saying that low INP concentrations compared to N_ice concentrations are observed previously, but this response does not answer the above question. There are no results regarding the nature of INPs or the freezing spectra of droplets at colder temperatures to understand this concern. My all other questions are somewhat related to this concern. Additional experimental evidence (for example as above) is needed to support the claims made in the paper.

In additional experiments we certainly would find residues or foreign substances in the planar branched crystals we categorise as secondary ice. Such residues could be cloud condensation nuclei in rime droplets, scavenged interstitial aerosol particles, or others. Some of these residues may indeed be capable of triggering ice at temperatures colder than -25 °C. However, initial ice formation at such cold temperatures would not have resulted in the form (habit) of crystals we have analysed. For this reason, we are convinced that they resulted from an ice multiplication process. There is strong evidence supporting this view, which we would include in a revised version of the manuscript:

The temperature range from -20 °C to -70 °C is the so-called "polycrystalline regime" dominated by crystal shapes with a range of different angles between branches or plates extending in three dimensions (Bailey and Hallett, 2009). These crystals will continue to grow when falling into warmer layers of air, as long as these layers are supersaturated with respect to ice. Otherwise, the crystals will sublimate. The growth habit of the falling crystals may change depending on temperature and supersaturation, but it will remain polycrystalline and irregular (c.f. Fig. 6 and 7 in Bailey and Hallett, 2009). Polycrystalline ice particles are highly unlikely to grow into the kind of crystals we have sampled, which had the same angle (60°) between all branches, and branches only extending in a single plane (i.e. dendrites; c.f. Schwarzenboeck et al., 2009).

We have moved the sentence mentioned by the referee from the introduction section to the results and discussion section.

The lack of INP$_{-17}$ indicates that the formation of these crystals was most likely not triggered by heterogeneous freezing, but through a secondary ice formation process. (page 7, lines 4-5)

We have added the second paragraph in our comment above to the results and discussion selection (see page 8 lines 7-13 in the revised version of the manuscript).

We think that dendritic crystals either formed through primary or secondary ice formation processes are initially formed at around -15 °C, a temperature which is encoded in their habit. Only around such a temperature these habits are formed, which was shown by many studies cited in our manuscript. We think that formation of these crystals can not have been triggered by INPs that were activated in the atmosphere below -20°C, otherwise they would have grown into polycrystalline, not single planar crystals. Therefore, we are convinced that the freezing of melted crystal droplets below -20 °C was caused by CCN scavenged during rime formation on the dendrites. Some CCN that had not been activated as INP before or during collision with the dendrites, will have been activated as INP on our cold stage when its temperature dropped below -20 °C and caused the test droplets to freeze.

[revised manuscript text omitted]

---

## Referee Report (RR1)

**New type of evidence for secondary ice formation at around -15°C in mixed-phase clouds**

**General comments**

The authors are presenting a method to assess the likelihood of secondary ice production on a per-hydrometeor basis. They have been thorough in setting up the new experimental apparatus and have used it over a month at Jungfraujoch. The authors note that the setup is "field deployable", so that it could be used also in field campaigns. My specific comments have mostly been addressed, but given that the novelty is in the methodology, I wonder if Aerosol Measurement Techniques would not be the better fit for this manuscript. The scientific conclusions still seem limited to me. For example, it is a bit extreme to state that "no conclusion regarding the process of secondary ice formation can be drawn from our observation." Could not the meteorological data be used at least to speculate on more and less likely secondary mechanisms? Is the enhancement factor higher if the cloud base temperatures or horizontal winds are stronger? Or if the winds come from one direction or another?

I also want to say that I still have reservations about the ability of this method to estimate ice enhancement factors for mixed-phase clouds in general. Were all (or almost all) dendritic ice crystals retained from the flow across the black aluminum plate during sampling periods? If so, it is impressive that there were only 229 such crystals over 10 days. If not, representativeness is still a concern. The authors state that "if we had the crystals from a small fraction of a cloud volume and would extrapolate our findings to a much larger volume in which primary and secondary crystals are very heterogeneously distributed, we would face a problem." But as I understand it, this is what is being done. It is stated very generally in their responses that they "can draw a conclusion regarding secondary ice formation within mixed-phase clouds".

Let us set aside this concern because it is still interesting to look at individual ice crystals. Some caution needs to be taken in any discussion of ice crystal habit and ice formation: ice crystal habit encodes information about *growth temperature* not *formation temperature*. Ice crystallization is a kinetic process and dependent on the crystal's temperature-supersaturation history. It is possible to nucleate at a cold temperature and then enter a warmer temperature zone – by sedimentation, advection, etc. – and do most of the growth there. It seems unlikely to me that homogeneously nucleated ice crystals move into a zone of -15°C before significant growth has occurred, so that the method should generally not have false positives in this way. But I do think that this kinetic nature of ice crystallization warrants mention within the manuscript.

I appreciate that photographs of ice crystals have been added. Those in the supplemental material, and in fact all of the text and imagery in the supplement, could be added to the main manuscript in my opinion. This is again given the emphasis on a new technique. Finally, given that "closer inspection of the enlarged photographs" indicated that some were not planar or branched, it would be nice to have a more rigorous means of classification for future studies. Would there be a way to use the ImageJ software used for sizing to also do some kind of "shape processing"? If the authors have ideas for rigorous classification algorithms, they could mention these within the conclusion section. I have only a few other specific comments.

**Specific comments**

Page 2, Line 13-14 – This point is slightly confusing (because secondary ice is associated with warmer temperatures and here you are mentioning colder temperatures). I would rewrite as *Because they all (n = 301) re-froze only at temperatures substantially lower than the measured cloud top temperature, the authors presumed them to be of secondary origin.*

Page 3, Line 19-24 – In my opinion, it makes more sense to list the motivations to focus on -15°C in a different order. This is a detail, but the first motivation is really the distinctive ice habit at this temperature. Thereafter, the crystals have lower density and terminal velocity, and the observations of Westbrook and Illingworth (2031) and the higher ice-ice collisional efficiency seem reasonable.

Page 3, Line 30 – "nucleated" not "catalysed"

Section 2.2 – My former concern about INP coagulation and sedimentation within the larger volume droplet was not addressed. It is favorable that "*the procedure takes ~15 minutes*", but there is still sufficient time for a non-negligible drop in particle surface area (see Emersic et al. 2015 ACP Figure 8). This caveat needs to be mentioned.

Page 4, Line 18-19 – How are you able to "*exclude hoar frost particles*"?

Page 4, Line 25 – How exactly were the "*images … later analysed more exactly*"? Visually?

Table 1 – Thank you adding the standard deviations.

---

## Author Response (AR2)

**Point-by-point explanation of the changes made to the manuscript in response to the comments received during the closed review process**

First of all, we would like to all the reviewers during the process. Special thanks to Sylvia C. Sullivan for her valuable comments during the whole review process as well as the fresh evaluation of the anonymous referee.

For clarity and easy visualization, the referee's comments are shown from here on in black.

> The authors' replies are in blue font with an increased indent below each of the referee's statements.

>> The authors' comments about the changes made to the manuscript after the first review round are stated in green, with a further increased indent.

>> Furthermore, the relevant changed sections from the revised manuscript are copied below in red. Page and line numbers (in red) refer to the revised version of the manuscript (without track changes).

**Authors' response to Referee #1**
**Review from Sylvia C. Sullivan received: 27 November 2018**

General comments

The authors are presenting a method to assess the likelihood of secondary ice production on a per-hydrometeor basis. They have been thorough in setting up the new experimental apparatus and have used it over a month at Jungfraujoch. The authors note that the setup is "field deployable", so that it could be used also in field campaigns. My specific comments have mostly been addressed, but given that the novelty is in the methodology, I wonder if Aerosol Measurement Techniques would not be the better fit for this manuscript. The scientific conclusions still seem limited to me. For example, it is a bit extreme to state that "no conclusion regarding the process of secondary ice formation can be drawn from our observation." Could not the meteorological data be used at least to speculate on more and less likely secondary mechanisms? Is the enhancement factor higher if the cloud base temperatures or horizontal winds are stronger? Or if the winds come from one direction or another?

> We think that ACP is fitting well because the focus of the manuscript is more on results of the new methodology than about the technique itself.

> Indeed, we can elaborate our scientific conclusion by speculating about the likely mechanisms of secondary ice production. However, the scope of our speculation is limited by the low throughput of our method, which does not allow to resolve with the necessary precision possible short-term fluctuations in ice multiplication due to changes in meteorological conditions between individual winter storms at Jungfraujoch during that observation period.

>> We have speculated on the mechanisms that could have played a role in secondary ice formation. With the help of the estimated cloud base temperature and the findings by Sullivan et al. (2018) we think that droplet shattering is less likely than rime splintering and ice-ice collision breakup. We further included a suggestion by the second, anonymous referee.

Because the estimated cloud base temperature was mostly below 0 °C during our observations, rime splintering and ice-ice collision breakup are more likely to have played a relevant role as secondary ice formation processes, as compared to droplet shattering (Sullivan et al., 2018). Whichever process was operating, it must have produced very small fragments, otherwise there would not have grown singular, regular, branched crystals (e.g. dendrites) from them (page 9, lines 19-24)

However, it is not possible to make a statement like: "higher ice enhancement factor correlates with stronger wind speeds". The standard deviation of the ice multiplication factor for a single day is relatively large. Furthermore, we have only a dataset of 10 days.

We have clarified that the low throughput of the method only provides for averaging over prolonged sampling periods and not for investigation of single clouds in the discussion and in the conclusion.

[…] but we can not make detailed judgements about single clouds. (page 7, line 23)

The low throughput only provides for averaging over prolonged sampling periods and not for investigating single clouds. (page 9, line 16-17)

Nevertheless, resolving relations between meteorological conditions and ice multiplication may be possible on a different (spatial) scale with our approach. We have added this point to our conclusion by including the finding by Phillips et al. (2017).

There are locations or meteorological weather conditions with dendrites that are less rimed. It would be interesting to repeat the measurements for such conditions. We would expect a lower ice-ice collision breakup efficiency if the dendrites are less rimed, at least if ice-ice collision breakup would play a role.

[…] or where they are less rimed. Less riming is likely to generate a smaller number of fragments by ice-ice collision breakup of dendrites as parametrized by Phillips et al. (2017). Under such conditions we would expect to find a smaller ice multiplication factor. (page 9, lines 25-27)

I also want to say that I still have reservations about the ability of this method to estimate ice enhancement factors for mixed-phase clouds in general. Were all (or almost all) dendritic ice crystals retained from the flow across the black aluminum plate during sampling periods? If so, it is impressive that there were only 229 such crystals over 10 days. If not, representativeness is still a concern. The authors state that "if we had the crystals from a small fraction of a cloud volume and would extrapolate our findings to a much larger volume in which primary and secondary crystals are very heterogeneously distributed, we would face a problem." But as I understand it, this is what is being done. It is stated very generally in their responses that they "can draw a conclusion regarding secondary ice formation within mixed-phase clouds".

Not all the dendritic ice crystals collected on the plate were analysed. We have randomly sampled ice crystals from different clouds and different days. If there were dendrites after briefly exposing the plate to the precipitating cloud, we analysed two of them and then had at least a gap of ~15 minutes before the next collection. Therefore, we think that our sampling procedure resulted in a representative mix of crystals from the full range of mixed-phase clouds during the observations at Jungfraujoch. The results might not be representative for a single cloud specifically because of the low throughput, but they represent an average for the entire campaign.

We have added a sentence in the method section about the sampling technique.

Generally, we exposed the plate for some seconds to the precipitating cloud until at least two dendritic snow crystals had deposited on it and then analysed those. (page 4, line 18-19)

Furthermore, we have changed slightly the paragraph on page 7 from line 18-25 to be more precise and added the following sentence:

Since we have randomly sampled crystals from many clouds over a prolonged period, we can extrapolate the found multiplication factor to dendrites in MPCs at Jungfraujoch during winter months in 2018 […] (page 7, lines 21-24)

Let us set aside this concern because it is still interesting to look at individual ice crystals. Some caution needs to be taken in any discussion of ice crystal habit and ice formation: ice crystal habit encodes information about growth temperature not formation temperature. Ice crystallization is a kinetic process and dependent on the crystal's temperature-supersaturation history. It is possible to nucleate at a cold temperature and then enter a warmer temperature zone – by sedimentation, advection, etc. – and do most of the growth there. It seems unlikely to me that homogeneously nucleated ice crystals move into a zone of -15°C before significant growth has occurred, so that the method should generally not have false positives in this way. But I do think that this kinetic nature of ice crystallization warrants mention within the manuscript.

We will point out in the discussion that the ice crystal habit encodes information about the growth temperature and not the formation temperature, and that it seems unlikely that homogenously nucleated ice crystals could have moved into a zone of -15 °C to then grow into single crystals (e.g. dendrites).

We have added a few sentences along the manuscript to carefully point out the differences between both temperatures. Furthermore, we have added a sentence including the reference by Furukawa (1982) to mention that it is highly unlikely that dendritic crystals were grown from initial ice crystal that had been formed by homogeneous freezing.

Assuming that the growth temperature of a crystal is not much different from the temperature of its initial formation, these observations suggest that […] (page 3, line 26-27)

It is highly unlikely that these crystals had grown from homogenously frozen cloud droplets. Homogenous freezing at a temperature well below -20 °C results in a polycrystalline initial ice crystal from which a polycrystalline snow crystal develops (Furukawa, 1982), and not a single crystal like a dendrite. (page 7 lines 10-13)

The ice crystal habits encode information about the growth temperature of the crystals not their formation temperature. The growth temperature […] (page 8 line 15-16)

Based on these findings, the information of growth temperature encoded in the habit of a crystal enables an assumption about the temperature range at which the crystal formed. For dendritic crystals, we can assume that the initial formation temperature is likely above -20 °C. (page 8, lines 29-31)

I appreciate that photographs of ice crystals have been added. Those in the supplemental material, and in fact all of the text and imagery in the supplement, could be added to the main manuscript in my opinion. This is again given the emphasis on a new technique. Finally, given that "closer inspection of the enlarged photographs" indicated that some were not planar or branched, it would be nice to have a more rigorous means of classification for future studies. Would there be a way to use the ImageJ software used for sizing to also do some kind of "shape processing"? If the authors have ideas for rigorous classification algorithms, they could mention these within the conclusion section. I have only a few other specific comments.

We are happy to hear so. We will leave the supplemental material in the supplement because it is not necessarily needed to validate our conclusions. We will change the mentioned sentence indicating how we classified the crystals. We have visually classified the crystals. Machine learning tools exist which classify the crystals automatically into different categories, like for instance developed by Praz et al. (2017), but their classifications are currently not as differentiated as we would need for the purpose of our study.

We have added that we have visually classified the crystals and we have pointed towards Praz et al (2017), which is a possibility to classify the crystals more rigorously.

Images were later analysed visually and not by machine learning methods, such as developed by Praz et al. (2017), […] (page 4 line 27-28)

Specific comments

Page 2, Line 13-14 – This point is slightly confusing (because secondary ice is associated with warmer temperatures and here you are mentioning colder temperatures). I would rewrite as Because they all (n = 301) re-froze only at temperatures substantially lower than the measured cloud top temperature, the authors presumed them to be of secondary origin.

Thank you, we will do so.

We have rewritten it almost word by word as suggested.

Because they all (n = 301) re-froze only at temperatures substantially lower than the estimated cloud top temperature, the authors presumed them to be of secondary origin. (page 3 line 12-14)

Page 3, Line 19-24 – In my opinion, it makes more sense to list the motivations to focus on -15°C in a different order. This is a detail, but the first motivation is really the distinctive ice habit at this temperature. Thereafter, the crystals have lower density and terminal velocity, and the observations of Westbrook and Illingworth (2031) and the higher ice-ice collisional efficiency seem reasonable.

We will do so.

We have changed the orders as suggested and changed formulations very slightly.

First, the growth habit of ice crystals forming in super-saturated conditions between -12 °C and -17 °C is well and distinctively defined. These are single, planar, branched, sector-type or dendrite-type habits (Nakaya, 1954; Magono, 1962; Magono and Lee, 1966; Takahashi et al., 1991; Takahashi, 2014; Libbrecht, 2017) that grow by vapour diffusional growth into a diameter of several millimetres during a vertical fall of a few 100 m (Fukuta and Takahashi, 1999). Second, Westbrook and Illingworth (2013)

observed a long-lived supercooled cloud layer with a cloud top temperature around
-13.5 °C, which continued to precipitate ice crystals well beyond the expected
exhaustion of its INP reservoir. Third, laboratory investigations revealed ice-ice
collision to be most effective in producing secondary ice particles at around -16 °C
(Takahashi et al., 1995), or in collisions involving dendritic crystals (Griggs and
Choularton, 1986). (page 3, lines 18- 26)

Page 3, Line 30 – "nucleated" not "catalysed"

Changed.

Section 2.2 – My former concern about INP coagulation and sedimentation within the larger volume
droplet was not addressed. It is favorable that "the procedure takes ~15 minutes", but there is still
sufficient time for a non-negligible drop in particle surface area (see Emersic et al. 2015 ACP Figure
8). This caveat needs to be mentioned.

We have mentioned it.

In total, the procedure (i.e. collecting and analysing two samples) takes ~15 minutes,
a time interval which may allow for a reduction in particle surface area due to
coagulation (Emerstic et al., 2015). (page 5, line 28-30)

Page 4, Line 18-19 – How are you able to "exclude hoar frost particles"?

We exclude small and irregular ice crystals

We have changed our phrasing.

Our selection criteria excluded small or irregular ice crystals, which are more typical
for hoar frost particles which might have been generated by local surface sources
around the station (Llyod et al., 2015; Farrington et al., 2016; Beck et al., 2018).
(page 4, line 19-21)

Page 4, Line 25 – How exactly were the "images … later analysed more exactly"? Visually?

Yes, we have analysed them visually. (Already mentioned in one of the answers below.)

We have changed this.

Images were later analysed visually and not by machine learning methods, such as
developed by Praz et al. (2017), […] (page 4 line 27-28)

Table 1 – Thank you adding the standard deviations.

You are welcome.

Authors' response to Referee #3

**Review from the anonymous referee received: 29 November 2018**

I support publication. See below for a more complete explanation. An overview of how I reviewed
the paper is also warranted. I read the paper after reviews 1 and 2 (and the corresponding replies
from the authors) had already been posted. I read the paper first, so that I could form an
independent opinion, then read the other reviews and replies. Finally, I read the revised version of
the manuscript (and associated correspondence from the authors) before I wrote out my own
review.

The authors have addressed a particularly pernicious issue in cloud physics -- secondary ice
formation in a regime where riming-splintering is not occurring. As the authors have noted, field
observations indicate that there's more ice than can be explained with the measurements of ice
nuclei. There are hypotheses as to what processes produce this ice, but data is scarce.

This study is rare, in that the authors attempt to derive a multiplication factor from field data. While
there are uncertainties associated with their technique, it's quite intriguing and worthy of
publication. Using the ice crystals as a measure of the temperature is the key step in this procedure,
and I think that it is justified. The (pristine) habit is a good indicator of temperature and saturation
ratio. Using planar, branched crystals in mixed phase cloud conditions restricts the temperature
range to $\approx$ -15 C. By collecting individual crystals, melting them, then re-freezing them, the
authors establish whether there is an ice nucleating particle within the crystal capable of nucleating
ice at -15 or higher. The absence of such an entity is strong evidence that the crystal was formed
through a multiplication process. If the crystal is the result of a secondary process, and it collects
other interstitial aerosol particles or aerosol particles embedded in supercooled droplets (\textit{i.e}
through riming) before being sampled, those particles will most likely not be effective ice nucleating
particles for $T \geq -15$.

Those facts, taken together, are strong indicators of primary vs. secondary formation processes. The
habit indicates the temperature of formation, and the re-freezing temperature indicates the
presence or absence of a particle capable of catalyzing freezing at -15, which in turn is an indication
of primary vs. secondary formation.

Are there assumptions and uncertainties inherent in this method? Yes. The other reviewers have
pointed out several; the changes that the authors have made in the revised manuscript have
improved it substantially. \vskip .2cm

The key issue is whether the assumptions and uncertainties that are inherent in this technique are
so great as to preclude publication. I do not think they are. \vskip .2cm

This is just musing, and a not-quite-rhetorical question on my part... Does this data imply that the
secondary process here produces ice fragments that are quite small? If the process produced large
fragments, wouldn't those show up as irregularities in the collected crystals. (In other words, the
habits wouldn't be so pristine..?) To be clear, I'm not asking that the authors settle this question, just
posing it as something to consider.

Thank you very much for your review. We were pleased to read such a supportive
evaluation. Furthermore, we appreciated your question regarding the size of the ice
fragments.

We have integrated the answer to your question into the conclusion section by adding a sentence.

[revised manuscript text omitted]